# Basin-scale gyres and mesoscale eddies in large lakes: A novel procedure for their detection and characterization, assessed in Lake Geneva

Seyed Mahmood Hamze-Ziabari[1], Ulrich Lemmin[1], Frédéric Soulignac[1,2], Mehrshad Foroughan[1] and David Andrew Barry[1]

[1] Ecological Engineering Laboratory (ECOL), Environmental Engineering Institute (IIE), Faculty of Architecture, Civil and Environmental Engineering (ENAC), Ecole Polytechnique Fédérale de Lausanne (EPFL), 1015 Lausanne, Switzerland
[2] Commission Internationale pour la Protection des Eaux du Léman (CIPEL), Nyon, Switzerland

*Correspondence to*: Seyed Mahmood Hamze-Ziabari (mahmood.ziabari@epfl.ch)

**Abstract.** In large lakes subject to the Coriolis force, basin-scale gyres and mesoscale eddies, i.e., rotating coherent water masses, play a key role in spreading biochemical materials and energy throughout the lake. In order to assess the spatial and temporal extent of gyres and eddies, their dynamics and vertical structure, as well as to validate their prediction in numerical simulation results, detailed transect field observations are needed. However, at present it is difficult to forecast when and where such transect field observations should be taken. To overcome this problem, a novel procedure combining 3D numerical simulations, statistical analyses, and remote sensing data was developed that permits determination of the spatial and temporal patterns of basin-scale gyres during different seasons. The proposed gyre identification procedure consists of four steps: (i) data pre-processing, (ii) extracting dominant patterns using Empirical Orthogonal Function (EOF) analysis of Okubo-Weiss parameter fields, (iii) defining the 3D structure of the gyre, and (iv) finding the correlation between the dominant gyre pattern and environmental forcing. The efficiency and robustness of the proposed procedure was validated in Lake Geneva. For the first time in a lake, detailed field evidence of the existence of basin-scale gyres and (sub)mesoscale eddies was provided by data collected along transects whose locations were predetermined by the proposed procedure. The close correspondence between field observations and detailed numerical results further confirmed the validity of the model for capturing large-scale current circulations as well as (sub)mesoscale eddies. The results also indicated that the horizontal gyre motion is mainly determined by wind stress, whereas the vertical current structure, which is influenced by the gyre flow field, primarily depends on thermocline depth and strength. The procedure can be applied to other large lakes and can be extended to the interaction of biological-chemical-physical processes.

## 1 Introduction

Pressure on large lakes, as major freshwater resources, is increasing due to a combination of changing climate and human activities. To a large extent, lake water quality is controlled by the pathways along which nutrients, phytoplankton and contaminants are transported/redistributed on meso- or basin-scales (Ralph, 2002); in this context, gyres and eddies play an

important role. Basin-scale gyres are usually restricted in the vertical by the thermocline depth (Ishikawa et al., 2002; Ji and Jin, 2006), which is also subject to high spatial and temporal variability due to the presence of mesoscale or submesoscale circulations (Nõges and Kangro, 2005; Ostrovsky and Sukenik, 2008; Corman et al., 2010).

Understanding gyre dynamics in large lakes has mainly been advanced by Csanady's theoretical analyses (Csanady, 1973, 1975). Wind forcing (the primary driver), the Earth's rotation (Coriolis force) and lakebed morphology are the main controlling parameters (Birchfield, 1967; Rao and Murty, 1970; Csanady, 1973; Pickett and Rao, 1977; Rueda et al., 2005; Shimizu et al., 2007; Nakayama et al., 2014). In order to provide and accumulate enough energy to generate basin-scale circulations such as gyres, wind with a relatively constant direction must blow over most of the water surface for a certain time (Csanady, 1975). It has been shown that in the Laurentian Great Lakes in North America, a downwind flow in shallower nearshore regions and an upwind return flow in the deeper part of a lake basin enhance the formation of such circulation cells. Stratification and the width of downwind flow, which is also known as a "coastal jet," will increase and confine the flow to the surface layer (Bennett, 1974). However, these studies were conducted under simplified boundary conditions, using bathymetry and atmospheric forcing data with limited resolution.

Theoretical studies have indicated that the response of a depth-variable lake to a strong uniform wind often leads to the formation of two counter-rotating gyres, also known as a double-gyre or dipole (Csanady, 1973; Bennett, 1974; Shilo et al., 2007), as was confirmed by early numerical simulations (Simons, 1980). With increased computational power, three-dimensional (3D) hydrodynamic models with higher accuracy, stability and resolution have since been developed and have significantly contributed to the understanding of lake circulation, especially that driven by wind (e.g., Beletsky et al., 1999; Laval et al., 2005; Beletsky and Schwab, 2008; Bai et al., 2013; Mao and Xia, 2020; Baracchini et al., 2020; Lin et al. 2022; Wu et al. 2022). Recent numerical simulations have also highlighted the role of baroclinicity induced by gradients of surface buoyancy (mainly surface heating and cooling), which can enhance the large-scale circulation in the Great Lakes (Schwab and Beletsky, 2003; Bennington et al., 2010; Verburg et al., 2011; McKinney et al., 2012).

Based on long-term current data of the Laurentian Great Lakes from a limited number of moorings, Beletsky et al. (1999) suggested that during the non-stratified season, a single counterclockwise rotating gyre exists in larger lakes (Lake Huron, Lake Michigan, Lake Superior), whereas a two-gyre pattern is established in smaller ones (Lake Ontario, Lake Erie). Stratification can modify this pattern. A two-gyre pattern was also observed in Lake Okeechobee (USA), a shallow (3.2 m mean depth) large lake (Ji and Jin, 2006). In Lake Biwa (Japan), topography, stratification and non-linearity were found to affect gyre development and could result in a three-gyre pattern (Akitomo et al., 2004). Determining the direct influence of atmospheric forcing on individual large-scale current systems in large lakes is challenging due to the complexity and potential range of hydrodynamic responses.

In parallel to increased computational capacity, high-resolution satellite images allow direct observations of gyres and mesoscale and submesoscale eddies (Steissberg et al., 2005; Zhan et al., 2014). In particular, Synthetic Aperture Radar (SAR) imagery can detect and characterize eddies in oceans (e.g., Johannessen et al., 1996; DiGiacomo and Holt, 2001;

Marmorino et al., 2010; Qazi et al., 2014). Although small coastal cyclonic eddies have been identified in Lake Superior with SAR imagery (McKinney et al., 2012), it has yet to be used to detect basin-scale gyres in lakes.

Numerical simulations of large-scale motions in Lake Geneva (e.g., Lemmin et al., 2005; Umlauf and Lemmin, 2005; Perroud et al., 2009; Le Thi et al., 2012; Razmi et al., 2013; Cimatoribus et al., 2018, 2019; Baracchini et al., 2020; Reiss et al., 2020) show that the Coriolis force is important in the force balance, and that gyres form. Although the existence of different gyre systems was suggested by these studies, none were confirmed with detailed field measurements. Based on long-term mooring data, albeit limited, Lemmin and D'Adamo (1996) suggested the existence of a basin-scale gyre. The formation of smaller scale eddies in two embayments was observed, affected by the embayment geometry (Razmi et al., 2013) and by meteorological conditions (Razmi et al., 2017).

In previous studies, numerical modeling results and observations from moored current meters were analyzed for the presence of large-scale gyres mainly by studying individual events. However, since field observations in large lakes are generally sparse in time and space, often limited to a few moorings, they cannot provide a detailed description of large-scale gyre patterns (Beletsky et al., 2013; Hui et al., 2021) and, in particular, determine whether these patterns are "typical" and thus important in the long-term development of the lake flow system. Therefore, in order to improve the understanding of the general circulation in large lakes, an algorithm that allows identifying and tracking basin-scale and mesoscale water mass movements in a lake is needed. In this context, one of the most popular methods in oceanography to identify and track eddies from Sea Level Anomaly maps and numerical modeling results is based on the Okubu-Weiss (OW) parameter (e.g., Isern-Fontanet et al., 2004; Xiu et al., 2010; Chang and Oey, 2014). The OW parameter allows separation of flow fields into vorticity-dominated regions (gyres and eddies) and strain-dominated regions (the ambient flow field) (Okubo, 1970; Weiss, 1991).

In the present study, high-resolution 3D hydrodynamic model results from Lake Geneva are combined with the OW parameter and Empirical Orthogonal Function (EOF) analysis in a novel procedure with the following objectives:

- To detect large-scale coherent flow features in large lakes, in particular large-scale gyres and mesoscale eddies.
- To design strategies for field campaigns that can confirm the existence of these features and thus, for the first time, provide unambiguous detailed field evidence of basin-scale gyres and mesoscale eddies in lakes.
- To identify links between atmospheric forcing patterns and the dominant variability of basin-scale flow systems that can help to predict the occurrence and the lifetime of the dominant flow patterns.

The Supplementary Information (SI) section provides additional figures, tables, etc., denoted with prefix S.

## 2 Materials and Methods

### 2.1 Study site

Lake Geneva (*Lac Léman*), the largest lake in Western Europe, is a crescent-shaped pre-alpine lake, situated between Switzerland and France (Figure 1). It is composed of a large eastern basin, the *Grand Lac*, with a maximum depth of 309 m and a western shallow and narrow basin, the *Petit Lac*, with a maximum depth of nearly 70 m. The lake is approximately 70-

km long along its main axis, has a maximum width of 14 km, a surface area of 580 km$^2$ and a volume of 89 km$^3$. The lake is located between the Alps to the south and east, and the Jura to the northwest. Two strong, dominant wind fields, namely the *Bise* (coming from the northeast) and the *Vent* (coming from the southwest), are guided by the surrounding topography (Wanner and Furger, 1990; Lemmin and D'Adamo, 1996). The central and western parts of the lake can experience strong wind events, which may last from several hours to several days. The eastern part of the lake is sheltered from strong wind events by the surrounding high mountains (Rahaghi et al., 2018; Lemmin, 2020).

The Coriolis force plays an important role in the formation of gyres, since the width of the lake is much larger than the internal Rossby radius of deformation; the typical range of the Rossby radius for Lake Geneva is O(5 km) during strongly stratified seasons, and O(1 km) during weakly stratified seasons (Lemmin and D'Adamo, 1996; Cimatoribus et al., 2018). Numerical simulations show that basin-scale gyres in Lake Geneva can rotate clockwise (anticyclonic) or counterclockwise (cyclonic) due to the surrounding topography and the two dominant wind fields, *Bise* and *Vent* (Figure 1a) (Razmi et al., 2017; Cimatoribus et al., 2018, 2019).

Lake Geneva is well suited for studying gyre dynamics and testing the new procedure since it exhibits the full range of hydrodynamic complexity expected in a large, oligomictic lake with a strong summer and weak winter stratification; full convective mixing only occurs during exceptionally cold winters. Furthermore, all background data needed for the development of this procedure are available in the public domain.

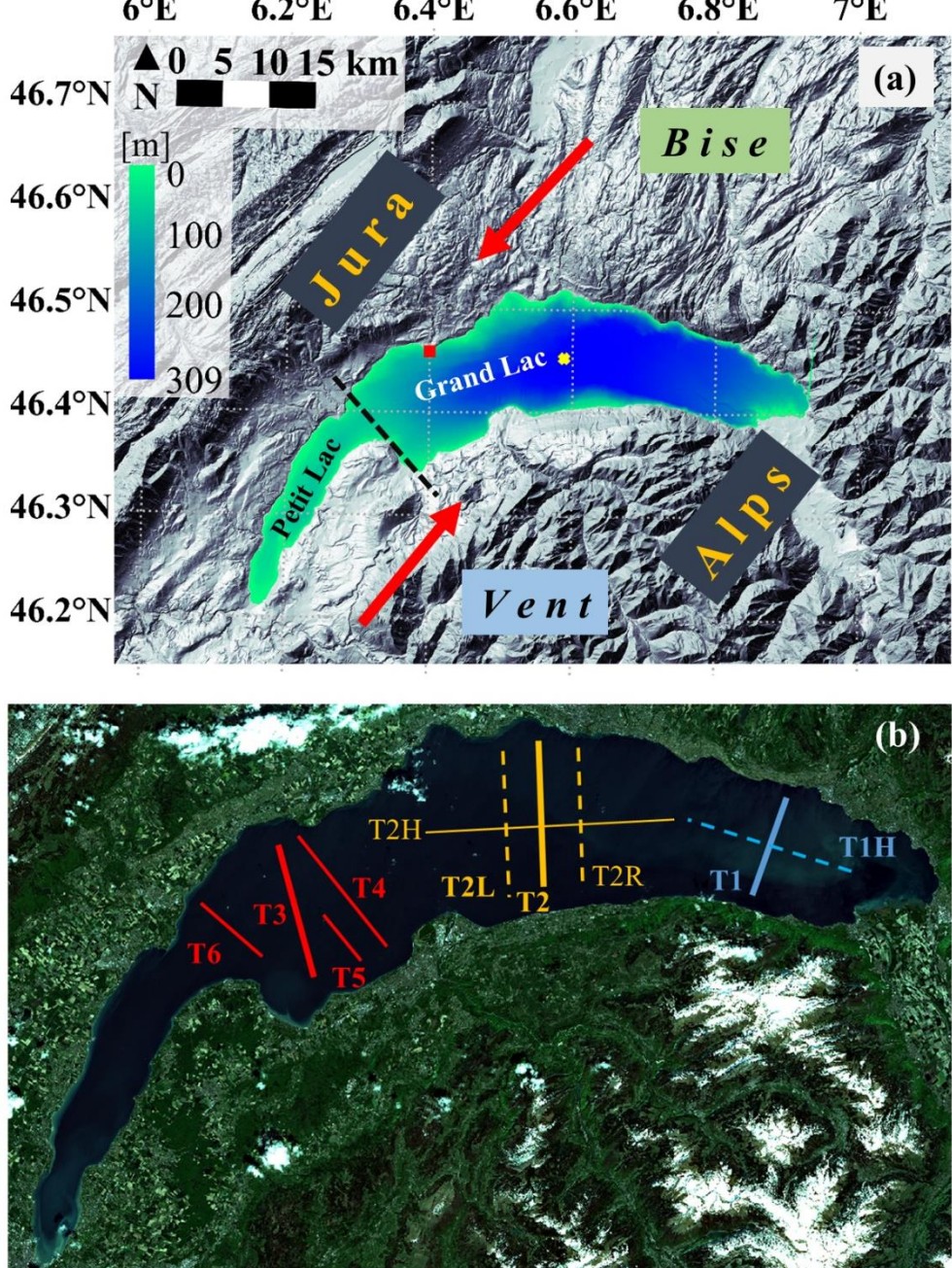

**Figure 1.** (a) Lake Geneva and surrounding topography, adapted from a public domain satellite image (NASA World Wind, last accessed 29 October 2022) and bathymetry data from SwissTopo (last accessed 2 August 2019). The colorbar indicates the water depth. SHL2 (yellow cross) is a long-term CIPEL monitoring station where data on physical and biological parameters are measured. The red square shows the location of the EPFL Buchillon meteorological station (100-m offshore). The black dashed line approximately delimits the two basins of Lake Geneva called the *Petit Lac* (west) and the *Grand Lac* (east). The thick red arrows indicate the direction of the two strong dominant

winds, called the *Bise* (coming from the northeast) and the *Vent* (coming from the southwest). (b) A schematic view of transects (T) selected for the different field campaigns in this study (for details, see text, and Table S1 in the Supplementary Information (SI) section). The background satellite image is taken from Sentinel-1A.

## 2.2 Hydrodynamic modeling

In this study, MITgcm (Marshall et al., 1997), which solves the 3D Boussinesq, hydrostatic Navier-Stokes equations (including the Coriolis force), was used in a series of numerical simulations for Lake Geneva for 2018 and 2019, based on the validated model setup of Cimatoribus et al. (2018). The model was forced by meteorological data (wind fields, atmospheric temperature, humidity, solar radiation) extracted from the COnsortium for Small-scale MOdeling (COSMO) atmospheric model of MeteoSwiss (last accessed 6 June 2021) (Voudouri et al., 2017). COSMO data will also be used in the Empirical Orthogonal Function (EOF) analysis. The first modeling step was a Low Resolution (LR) model (horizontal resolution 173 to 260 m, 35 depth layers, integration time step 20 s), which was initialized from rest using the temperature profile from CIPEL station SHL2 (CIPEL, 2019) measured on 25 October 2017 and 19 December 2018, respectively (calm weather conditions prevailed on both dates). For each run, the LR model spin up was ~180 d. The 3D interpolated results of the LR modeling were then used to initialize the High Resolution (HR) version of the model (horizontal resolution 113 m and 50 depth layers, with layer thicknesses ranging from 0.30 m at the surface to approximately 12 m for the deepest layer, integration time step of 6 s). For each year, the HR model was run for two seasons: under weakly stratified conditions and under strongly stratified conditions.

The model's capability to realistically reproduce stratification, mean flow, and internal seiche variability in Lake Geneva was demonstrated by Cimatoribus et al. (2018, 2019). To further assess the model's ability to reproduce seasonal stratification, a separate validation was carried out in the present study using temperature profiles measured in 2019 at SHL2 (Figure S1), located at the center of the *Grand Lac* (Figure 1a).

## 2.3 Transect field measurements

Based on the proposed procedure, six transects were selected in different parts of the *Grand Lac* (Figure 1b), i.e., one transect in the eastern part, and five in the central and western parts (details below). The large-scale gyre in the central part of the *Grand Lac* was the main focus of the transect field measurements due to its importance and persistence. Along each transect, ten profiles of current velocity spaced 1-km apart were measured, each for at least 5-10 min using an Acoustic Doppler Current Profiler (ADCP, Teledyne Marine Workhorse Sentinel) with the transducer located at 0.5-m depth. The ADCP was equipped with a bottom-tracking module, was set up for 100 1-m bins (blanking distance of 2 m), and operated in high-resolution processing mode. Tilt and heading angles were derived from sensors located inside the instrument. Vertical profiles of water temperature were measured with a Sea and Sun Marine Tech CTD75M multi-parameter probe at predefined points during the field campaigns. The Conductivity Temperature Depth (CTD) probe was lowered at a speed of ~10 cm s$^{-1}$. It recorded data at 7 Hz, thus providing a sampling resolution of ~1.5 cm.

**2.4 SAR remote sensing**

Synthetic Aperture Radar (SAR) is frequently used to detect oceanic surface features under light wind conditions (Johannessen
et al., 1996; Wang et al., 2019). The main advantages of SAR imagery are that: (i) it functions day and night under all weather
conditions, (ii) it has high sensitivity to small scale variability of the water surface, and (iii) it provides high resolution images.
The patterns observed in SAR images are due to the change of water surface roughness, which is influenced by wave/current
interactions, natural surface films and spatial variations of the local wind field (Johannessen et al., 2005). Gyres or eddies in
SAR images are indicated by dark spiral features called "black" or "classical" eddies (Karimova, 2012; Hamze-Ziabari et al.,
2022). More details are given in Text S1 in SI. For the present study, C-band SAR data were obtained from the European
Space Agency's (ESA) Sentinel-1A and Sentinel-1B satellites. The co-polarized VV (Vertical transmit, Vertical receive SAR
polarization) data were used because noise restricts the application of VH (Vertical transmit, Horizontal receive SAR
polarization) data (Gao et al., 2019). The spatial resolution of SAR data varies between 5 and 20 m for a ground sampling
distance of 10 m.

**2.5 Okubo-Weiss parameter and Empirical Orthogonal Functions**

The Okubo-Weiss (OW) parameter describes the local strain-vorticity balance in the horizontal flow field of a shallow fluid
layer. It allows separating vorticity-dominated regions associated with basin-scale gyres or mesoscale eddies from strain-
dominated ambient regions (Okubo, 1970; Weiss, 1991) and can be written as:

$$OW = S_n^2 + S_s^2 - \omega_z^2 \tag{1}$$

where $S_n$ is the normal component of strain, $S_s$ is the shear component of strain and $\omega_z$ is the relative vorticity of the flow,
defined by, respectively:

$$S_n = \frac{\partial u}{\partial x} - \frac{\partial v}{\partial y} \tag{2}$$

$$S_s = \frac{\partial v}{\partial x} + \frac{\partial u}{\partial y} \tag{3}$$

$$\omega_z = \frac{\partial v}{\partial x} - \frac{\partial u}{\partial y} \tag{4}$$

where $(u,v)$ are Cartesian components of the horizontal flow field. Positive OW values relate to strain-dominated regions,
whereas negative OW values identify vorticity-dominated regions.

For a given data set, Empirical Orthogonal Function (EOF) analysis identifies the main spatial patterns ($E$) and their
temporal evolution ($P$) (Wang and An, 2005). Hourly-averaged OW values derived from numerical simulations were
decomposed into the basis function $E_{OW}^k(X)$ and the principal component coefficient $P_{OW}^k(t)$ as:

$$OW(X, t) = \sum_{k=1}^{N} E_{OW}^k(X) P_{OW}^k(t) \tag{5}$$

where $k$ is the EOF mode, which varies from 1 to the maximum $N$, $X = (x,y)$ is the position under consideration and $t$ is time.
The $E_{OW}^k(X)$ modes identify the spatial patterns of the OW parameter. The time evolution of each $E_{OW}^k(X)$ mode can be
obtained from the time series of principal component coefficients, indicated as $P_{OW}^k(t)$.

## 3 Procedure for detecting gyres

The proposed gyre identification procedure consists of four steps (Figure 2): (i) data pre-processing, (ii) extracting dominant patterns using EOF analysis of OW fields, (iii) defining the 3D structure of the gyre, and (iv) finding the correlation between the dominant gyre pattern and environmental forcing.

In step (i), the OW values are computed for selected time windows using the horizontal velocity fields ($u$,$v$) generated by the 3D numerical modeling. Vorticity-dominated and strain-dominated regions can be identified from OW values, and can then, in step (ii), be used to locate and characterize the gyre flow field, in particular, gyre centers and the outer limits of gyres, located where vorticity and strain are approximately in balance. This is further supported/confirmed by patterns detected in SAR images. This OW/EOF analysis is repeated for several depths in step (iii) to verify that this surface pattern documents a gyre extending over the upper layers, with its depth limited by the thermocline. Step (iv) links the observed dominant OW pattern to the forcing that generated it, thereby identifying forcing patterns, mainly wind events, preceding the formation of gyre patterns. Scrutinizing actual meteorological data for such forcing patterns will then allow planning when and where to take detailed transect field measurements of the gyre pattern in the lake.

### 3.1 Detection of the core of a gyre or eddy

The separation of the OW field in terms of its sign can be used to detect cores of complex fluid flows (McWilliams, 1984; Elhmaïdi et al., 1993). These cores are characterized by negative OW values below a given (negative) threshold, $OW_T$ (Pasquero et al., 2001; Isern-Fontanet et al., 2006; Henson and Thomas, 2008; Liu et al., 2021). For this purpose, a threshold value of $OW_T = 0.2\sigma_{ow}$ was defined, where $\sigma_{ow}$ is the root-mean-square fluctuation of OW in the epilimnion with the same sign of the vorticity of the gyre/eddy core (Pasquero et al., 2001; Isern-Fontanet et al., 2006; Henson and Thomas, 2008).

In the data pre-processing stage, the hourly computed OW values were normalized by hourly values of $OW_T$ such that the normalized OW parameter, $OW_N$, partitions the topology of the gyre/eddy field into three regions: elliptic regions, $OW_N < -1$, hyperbolic regions, $OW_N > 1$, and a background field, $|OW_N| \leq 1$ (Pasquero et al., 2001; Isern-Fontanet et al., 2006). As shown in Figure 2, elliptic regions, which represent the center of gyres/eddies, are significantly more pronounced than the other regions. In lakes, basin-scale gyres are mainly restricted by the lake basin geometry and also impacted by the surrounding topography. Therefore, the regions with $OW_N > -1$ can be influenced by the interaction between gyres and nearshore boundaries. As a result, a wide spatial variability of $OW_N$ values in these regions was observed, whereas regions with $OW_N < -1$ were spatially rather stable. To eliminate such variability from the spatial and temporal identification of gyres/eddies, regions with $OW_N > -1$ were filtered out before implementing the EOF analysis.

For the proposed procedure (Figure 2), hourly filtered $OW_N$ values of different depth layers were computed for each month. The EOF analysis was then applied to detect the main modes (spatial pattern) and the corresponding principal component time series. From different modes of the EOF results, signatures of gyre or eddy cores were identified based on the following criteria:

1.  A local extreme value exists in the spatial modes of the $OW_N$ values
2.  At least one closed line exists around each local extreme value
3.  The core edge is identified as the closed line where the sign of $E_{OW_N}P_{OW_N}$ changes
4.  A vertical and horizontal coherence of a gyre or eddy signature must exist between different depth layers (bounded below by the thermocline), i.e., criteria 1-3 above must be spatially consistent for different depth layers.

By applying these criteria, the location of gyre and eddy centers can be identified.

## 3.2 Detecting the outer boundary of a gyre or an eddy

Gyre boundaries are located where vorticity and strain are approximately in balance, i.e., $|OW_N| \leq 1$. However, as discussed above, the outer gyre boundaries cannot be completely resolved by the OW analysis. Furthermore, the noise in the resulting $OW_N$ fields makes the detection of coherent flow structures difficult (Souza et al., 2011). Defining vertical and horizontal coherence criteria that can confirm coherent 3D gyre patterns can significantly reduce this limitation of the OW analysis. This threshold-based boundary detection method can be complemented by a geometry-based method using contour lines in Synthetic Aperture Radar (SAR) images. Therefore, information about the outer boundaries in support of the $OW_N$ analysis results can be obtained from SAR imagery, since basin-scale gyres and large eddies can be detected in SAR images.

## 3.3 Finding the link between the temporal variation of a gyre and environmental forcing

Wind stress and surface buoyancy flux (due to heating/cooling) are the processes controlling gyre circulations and their variability. To find a link between the pattern of external forcing and the computed spatial pattern of the $OW_N$ parameter presented in the previous sections, monthly EOF analyses of the total wind stress ($\sqrt{\tau_x^2 + \tau_y^2}$) and the net upward heat flux extracted from the COSMO atmospheric data are implemented. In the final stage of the procedure, the lagged cross-correlations between the spatial mode and the corresponding principal component time series of the environment forcing and the $OW_N$ are examined.

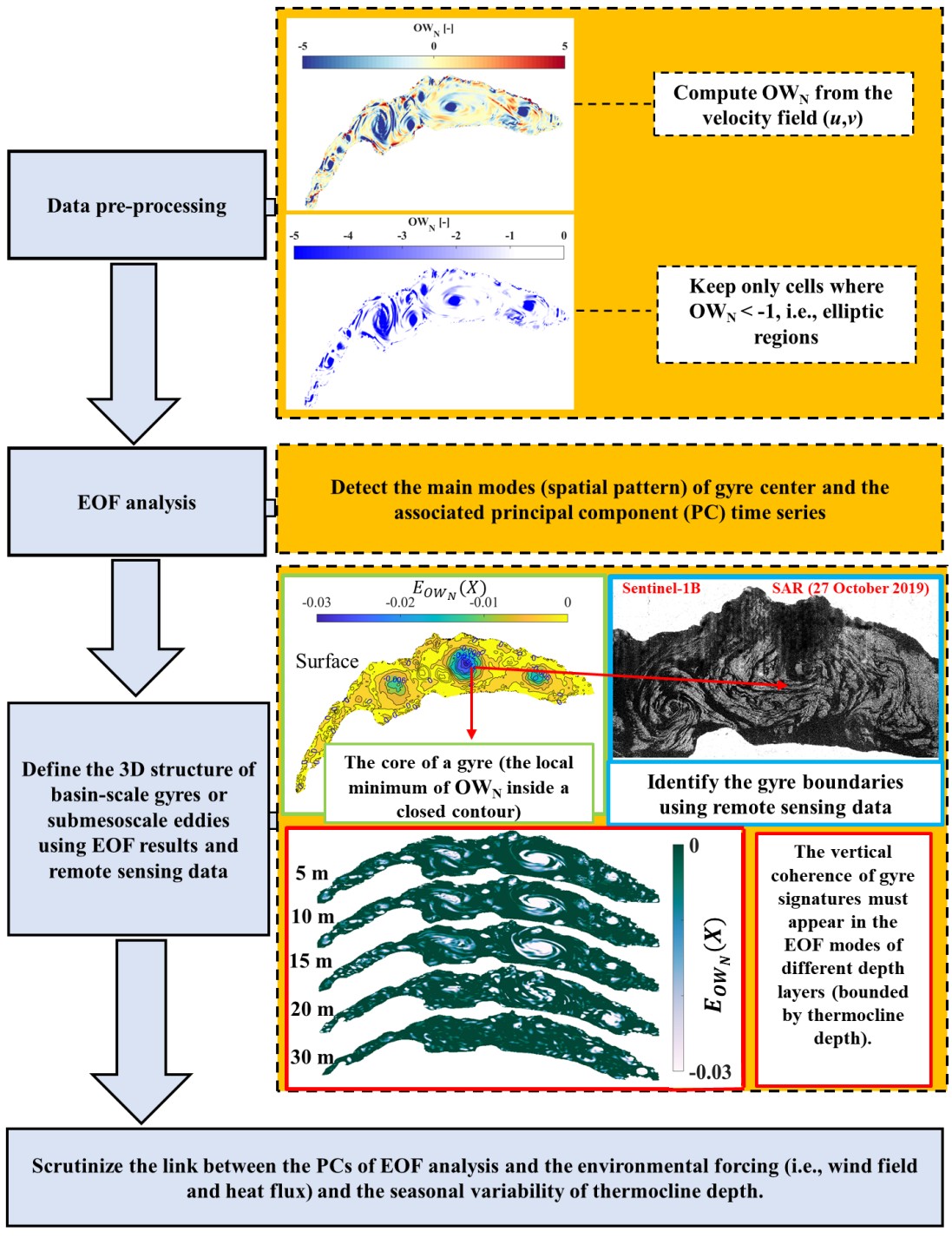

**Figure 2**. Flowchart of the proposed procedure showing the four steps (gray boxes) for detecting basin-scale gyres and eddies applied to Lake Geneva. Modeling results for the October 2019 *Bise* wind event, together with the corresponding SAR image, are used to highlight certain steps of the proposed procedure. For details, see text.

## 4 Results

In this section, the performance of the procedure for detecting the temporal and spatial variations of gyres/eddies is first evaluated for September 2018. Thereafter, the robustness of the proposed procedure is assessed by comparing results based on transect measurements from several transect field campaigns that were carried out based on the proposed procedure between September, when the lake is strongly stratified at a shallow depth and December, when it is weakly stratified at a greater depth.

### 4.1 Detecting gyres and eddies: Okubo-Weiss parameter and EOF analysis

#### 4.1.1 Detection of gyre center locations

Results of 3D modeling for September 2018 are used to demonstrate the performance of the proposed procedure. The first and second modes of the $OW_N$ variations in different depth layers (1, 5, 10, 15, 20 and 30 m) are given in Figures 3a,b. Due to the quick convergence of the EOF decomposition modes, only the first two dominant EOF modes are analyzed for detecting elliptic regions with negative $OW_N$ values. The first mode dominates the overall $OW_N$ variations. It accounts for nearly 56% of the total variance in the layers down to 15 m depth, and ~47% for deeper depths. The second mode contributes nearly 23 % to the total variance for the near surface layers. The remaining modes contain less than 5% of the total variance.

Three closed trajectories that encircle regions with negative $E_{OW_N}$ values can be distinguished in the first mode at different depth layers (Figure 3a). These regions indicate the presence of three large-scale gyres in the *Grand Lac,* if $E_{OW_N}P_{OW_N} < 0$ (the time series of $P_{OW_N}$ is discussed in the following sections). For better visualization, only negative values associated with $E_{OW_N}$, which document gyre patterns according to the definition in Section 3.1, are presented. Positive values of $E_{OW_N}$ are shown in Figure S2. The location of the center of each gyre in different depth layers is calculated by averaging the coordinates of the gyre centers (based on criteria 1-3 presented above). The closed trajectories weaken with depth down to 20 m and disappear at 30-m depth due to the presence of the thermocline, which is situated ~15-m depth.

A region bounded by closed trajectories with negative $OW_N$ values can also be considered as an indicator for areas where pelagic upwelling or downwelling are more likely to occur. Details of the spatial $OW_N$ pattern (e.g., Figure 4b, d) confirm the presence of negative $OW_N$ regions in the center of the large-scale gyres. The rotation sign of the three-gyre pattern after the *Bise* event is given in Figures S3-S5 for different months. The Anticyclonic Gyre (clockwise rotating) in the west and two Cyclonic Gyres (counterclockwise rotating) in the center and in the east of the *Grand Lac* are hereinafter referred to as AG, CG1 and CG2, respectively (Figure 3a). The second EOF mode reveals the simultaneous existence of smaller eddies, often with shorter lifetimes than basin-scale gyres (Figure 3b). More details on these eddies are presented below.

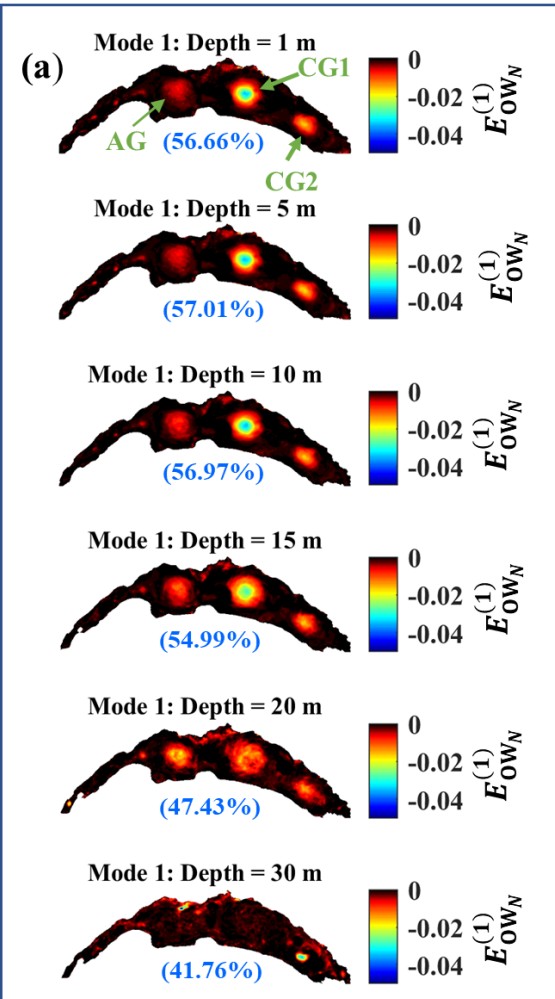 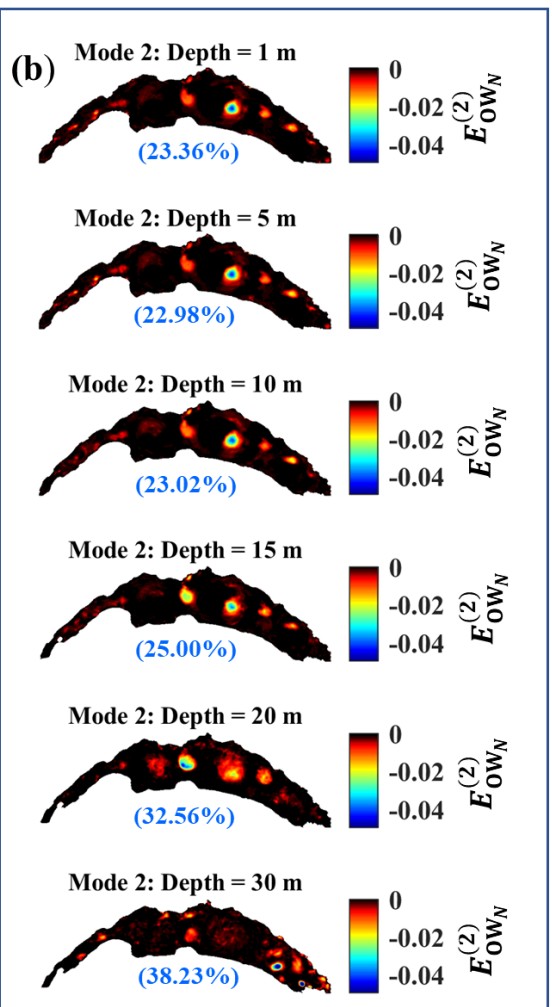

**Figure 3**. EOF analysis of the MITgcm output for Lake Geneva for the month of September 2018: (a) First mode (left column) and (b) second mode (right column) of the EOF of $OW_N$ are shown for different depth layers. The first mode is dominated by three large-scale gyres (circular zones of negative $E_{OW_N}$ values), whereas in the second mode, (sub)mesoscale eddies also appear. The contributions to the total variance of the monthly $OW_N$ values are given by the percentage indicated (in blue) for each layer and each mode. Colorbars give the range of the EOF.

### 4.1.2 Detection of the outer boundary of a gyre or an eddy

The $OW_N$ analysis shows that gyre centers have strongly negative $OW_N$ values and that positive $OW_N$ areas surround the gyres (Figure 4b, d). The transition zone between the two zones ($|OW_N| \leq 1$), which should indicate the outer edge of the gyre, is wide in certain parts of the circumference and does not allow determining the outer edge of the gyre (Figure 4b, d). SAR images may be used to confirm/define this edge (more details in Text S1). The boundaries between the two Cyclonic Gyres,

CG1 and CG2, located in the eastern part of lake, can be determined from SAR data obtained from Sentinel-1 on 21 July and 12 October 2018 (Figure 4a, c), where two (elliptical) gyres are evident. The minor/major axes of CG1 and CG2 are approximately 6.5/12.9 and 9.8/15.9 km, respectively. Smaller eddies, marked by strong negative $OW_N$ values in their center, surround the large-scale CG1 and CG2 gyres.

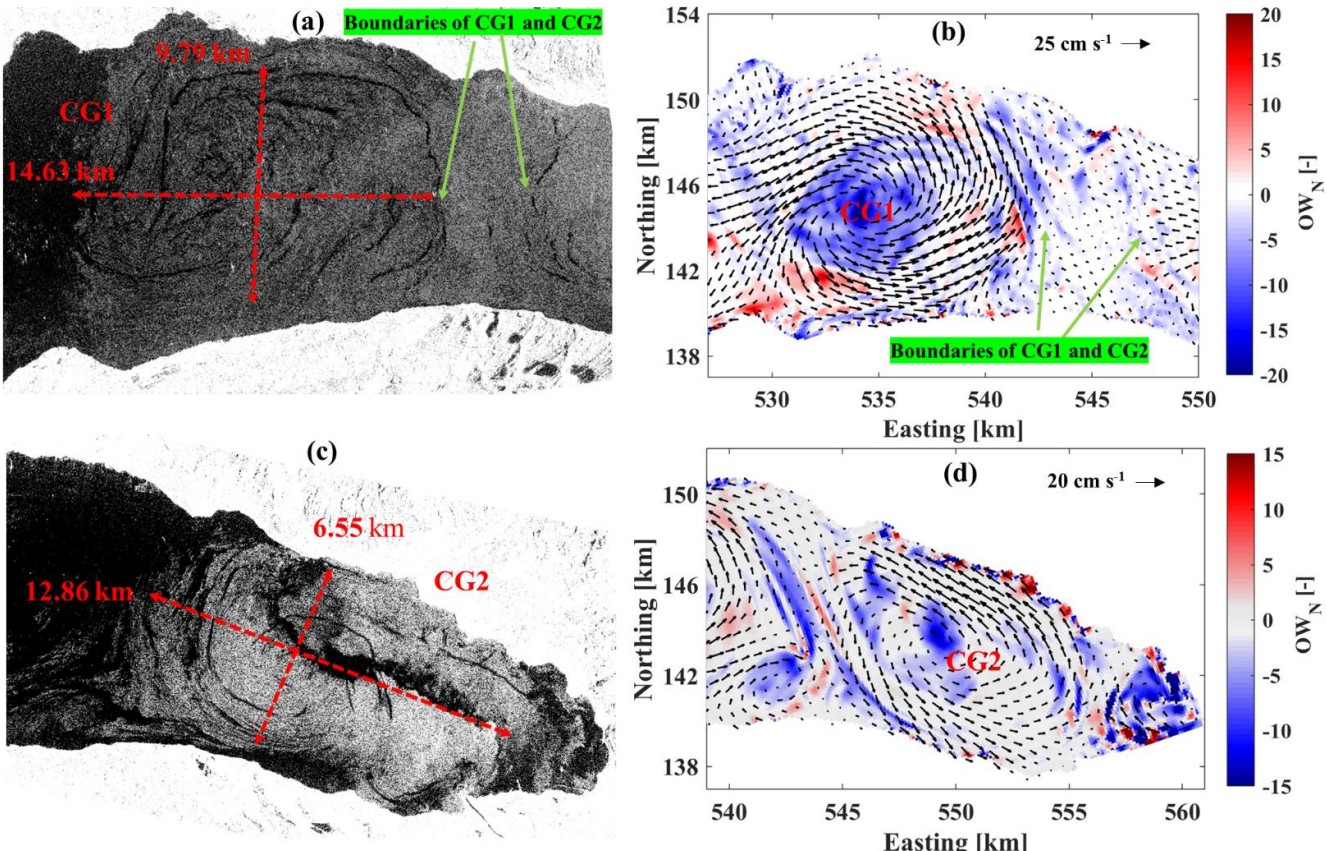

**Figure 4.** Left column: SAR images (Sentinel-1) indicating two Cyclonic Gyres (CG) in the eastern part of Lake Geneva for (a) CG1 (21 July 2018) and (c) CG2 (12 October 2018). Red dashed lines: major gyre axes and their dimensions. Right column: Corresponding modeled surface velocity fields (small black arrows) and $OW_N$ parameter values (colors). Strong negative $OW_N$ parameter values (blue) indicate the location of core zones of large-scale gyres and mesoscale eddies. The colorbars give the range of the $OW_N$ values.

### 4.1.3 Detecting large-scale gyres: Link between environmental forcing and gyre signature

To find a link between the pattern of external forcing and the computed spatial pattern of $OW_N$, EOF analyses of the total wind stress and the net upward heat flux extracted from the COSMO atmospheric data for September 2018 were carried out. The first spatial mode and the corresponding principal component time series of the total wind stress ($c_w^{(1)}(x)$, $\varphi_w^{(1)}(t)$) and the upward heat flux ($c_{HF}^{(1)}(t)$, $\varphi_{HF}^{(1)}(t)$) are presented in Figure 5a, b, respectively. The first mode of the external forces dominates

the variation of the forces, since it constitutes ~98% and 99% of the total variance related to the total wind stress and heat flux, respectively. The total variance differences between the first spatial modes and the principal component time series obtained from the solution with two dominant modes and the solution with all possible modes is negligible ($O(10^{-10}\text{-}10^{-14})$) compared to the calculated absolute values ($O(10^{-3})$) (Figure S6).

The EPFL Buchillon field station (Figure 1b) data show that a strong *Bise* event started at ~00:00 on 24 September 2018 and ended at ~11:00 on 26 September 2018. The mean wind speed ±1 standard deviation was $4.21 \pm 1.88$ m s$^{-1}$ with wind gusts of $8.40 \pm 3.37$ m s$^{-1}$. The mean wind direction was $61 \pm 13°$ (Figure 5e, f). Note that Buchillon field station data are only presented here to demonstrate that the EOF results shown in Figure 5a, b represent realistic wind fields; they are not used in the EOF analyses. Average wind speed and direction over the whole lake surface extracted from COSMO during the *Bise* event that lasted from 24 to 26 September 2018 are given in Figure S7. Figure 5e suggests that there is a link between the strong *Bise* event and the first EOF mode of OW$_N$ indicated by the principal component ($P_{\text{OW}_N}^{(1)}(t)$) time series. To determine this link, lagged cross-correlations were computed between the principal component time series of the dominant EOF modes of atmospheric forcing, i.e., the total wind stress ($\varphi_w^{(1)}(t)$) and the net upward heat flux, ($\varphi_{HF}^{(1)}(t)$), and the two dominant EOF modes of OW$_N$ (Figure 6). The lagged cross-correlation between the upward heat flux and $P_{\text{OW}_N}^{(1)}(t)$ does not exceed 0.3 in the depth layers influenced by the gyres (2-20 m; Figure 3). On the other hand, the cross correlation between $\varphi_w^{(1)}(t)$ and $P_{\text{OW}_N}^{(1)}$ reaches values > 0.8 in the same depth layers and a positive peak with a lag of 1-1.5 d for the near surface layers (1-20 m; Figure 6a-f). A positive peak means that $\varphi_w^{(1)}(t)$ and $P_{\text{OW}_N}^{(1)}(t)$ have the same sign. These results imply that the three-gyre pattern in the first mode (Figure. 3a) is predominantly controlled by the spatial $c_w^{(1)}(x)$ and temporal $\varphi_w^{(1)}(t)$ variations of wind stress. Furthermore, the time series of the first mode, $P_{\text{OW}_N}^{(1)}(t)$, in different depth layers suggest that the three-gyre pattern can persist for nearly 5 d after the wind peak (Figure 5e).

Thereafter (29 September 2018), the large-scale gyres break down into smaller gyres/eddies, and the second OW$_N$ mode dominates (Figure 5f). The cross-correlation between wind stress, upward heat flux and $P_{\text{OW}_N}^{(2)}(t)$ is not greater than 0.35 in the depth layers influenced by the gyres (Figure 6g-l). For wind stress, $E_{\text{OW}_N}^{(2)}$ is excited with a lag of ~5.5 d, with the same sign, and for upward heat flux with a lag of more than 4 d, again, with the same sign. Moreover, negative peaks in the cross-correlation between $E_{\text{OW}_N}^{(2)}$ and $\varphi_{HF}^{(1)}(t)$ show that $E_{\text{OW}_N}^{(2)}$ can be excited with a lag of 1-2 d due to differential heating and cooling during day-night cycles (Figure 6g-l). However, $E_{\text{OW}_N}^{(2)}$ is only weakly excited by wind stress or upward heat flux. In addition to the effects of environmental forcing presented above, the basin shape and bathymetry also change the gyre pattern over time.

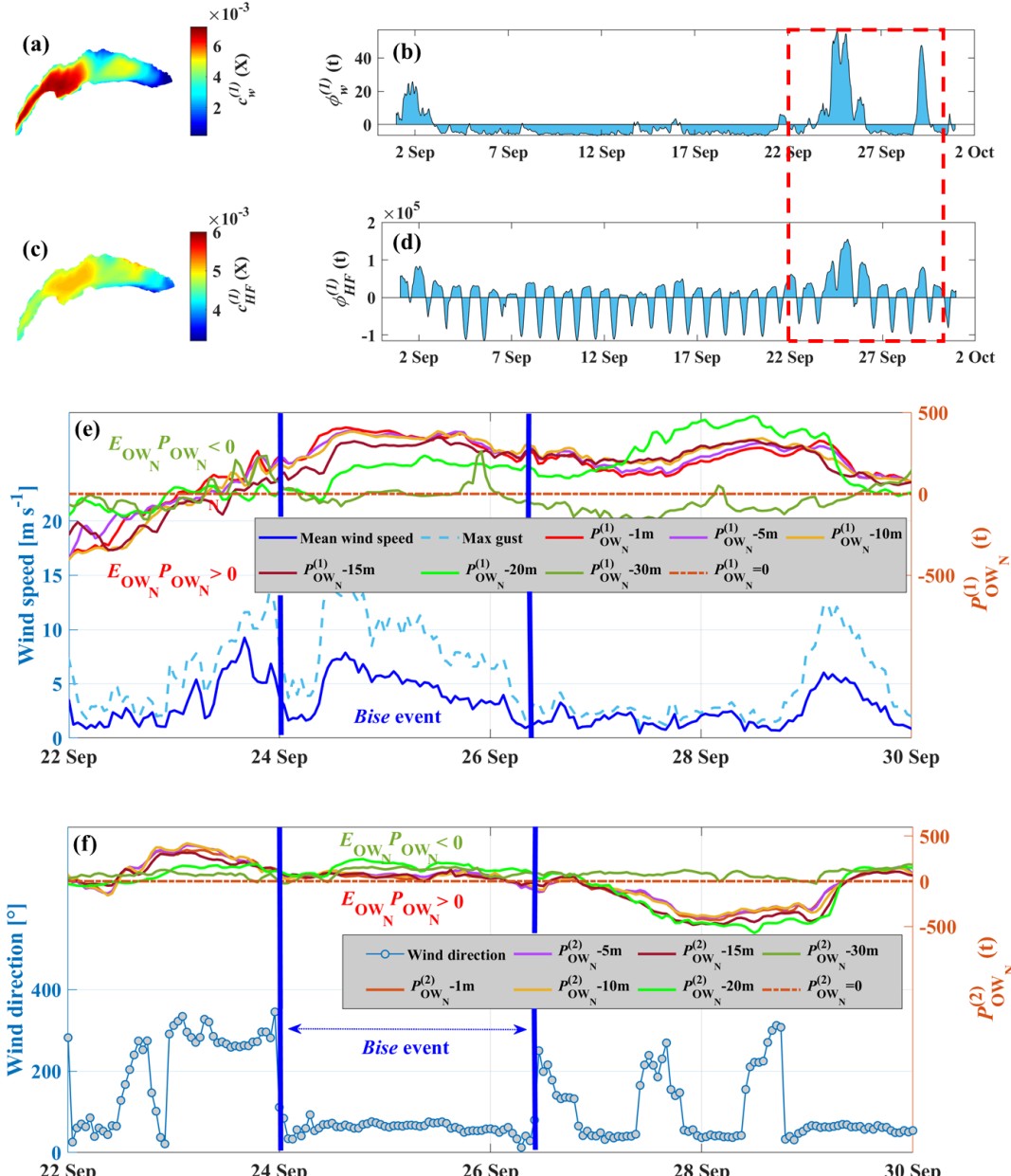

**Figure 5.** For September 2018: (a) First spatial mode ($c_w^{(1)}(X)$) and (b) principal component time series ($\varphi_w^{(1)}(t)$) of the total wind stress. (c) First spatial mode $c_{HF}^{(1)}(X)$ and (d) principal component time series ($\varphi_{HF}^{(1)}(t)$) of the net upward heat flux. (e) Top: Principal component time series of $OW_N$ associated with the first spatial mode ($P_{OW_N}^{(1)}(t)$) in different depth layers. Bottom: Mean wind speed and wind gusts at the Buchillon station (see Fig. 1 for location) for the period indicated by the red dashed-lined box in (b) and (d). (f) Top: The principal component time series of the $OW_N$ parameter associated with the second spatial mode ($P_{OW_N}^{(2)}(t)$) at different depth layers. Bottom: Wind direction at the Buchillon station for the period indicated by the red dashed-lined box in (b) and (d). Colorbars in (a) and (c) show the range

of the parameters. The colors in the legends of (e) and (f) correspond to the different depths. In (e) and (f), negative $(E_{OW_N} P_{OW_N})$ indicates the negative $OW_N$ values (the elliptic regions) that are characteristic of gyres. Note that all EOF analyses are based on COSMO meteo data. The two blue vertical lines mark the duration of the *Bise* wind event.

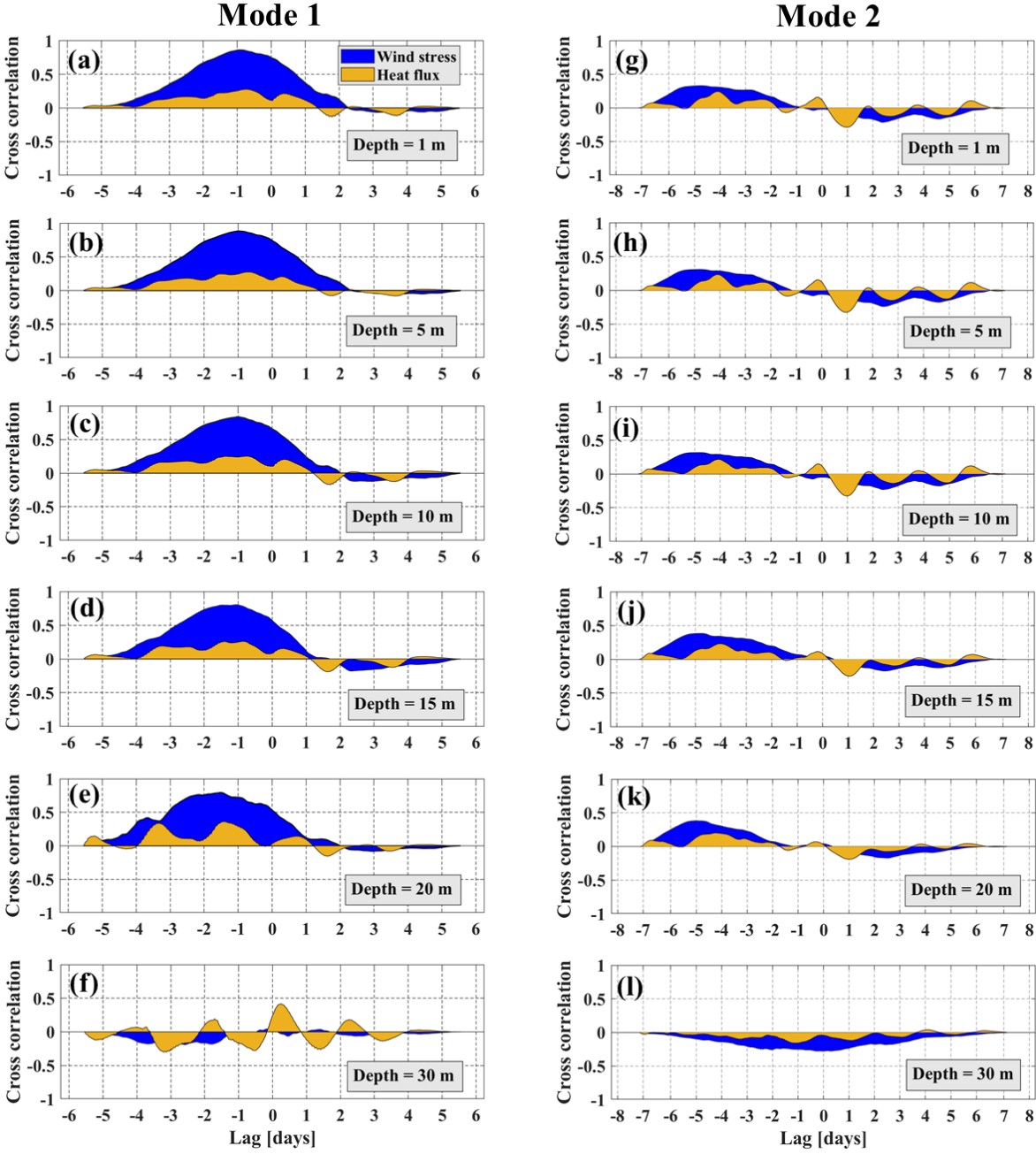

**Figure 6.** (a)-(f) Lagged cross-correlation of the principal component time series of the total wind stress ($\varphi_w^{(1)}(t)$) and the net upward heat flux ($\varphi_{HF}^{(1)}(t)$) with the principal component time series of the $OW_N$ parameter associated with the first spatial mode ($P_{OW_N}^{(1)}(t)$) in different

depth layers. (g)-(l) Lagged cross-correlation of the principal component time series of the total wind stress ($\varphi_w^{(1)}(t)$) and the net upward heat flux ($\varphi_{HF}^{(1)}(t)$) with the principal component time series of the $OW_N$ parameter associated with the second spatial mode ($P_{OW_N}^{(2)}(t)$) in different depth layers. Wind stress and heat flux are defined in color legend in panel (a).

## 4.2 Detecting large-scale gyres: Transect field campaigns

### 4.2.1 A cyclonic gyre at the center of the *Grand Lac*

To validate the proposed procedure for detecting the location and time of basin-scale gyres after a strong wind event, a field measurement campaign was performed in September 2019. This campaign focused on capturing the largest basin-scale gyre, CG1, in the central part of the *Grand Lac* (Figures 3, 4). Buchillon field data recorded a strong *Bise* event that started on 17 September 2019 at ~20:00 and ended on 20 September 2019 at ~13:00 (mean wind speed $\pm 1$ standard deviation was $3.65 \pm 0.86$ m s$^{-1}$, with gusts of $7.69 \pm 1.78$ m s$^{-1}$ and mean wind direction of $64 \pm 11°$). Furthermore, the spatial pattern of averaged

wind speed and direction computed from the forecasted COSMO atmospheric data (Figure 7a) revealed the same spatial pattern as had been observed for the September 2018 *Bise* event (Figure S7) discussed above. For September 2018, it was shown that the three-gyre system pattern is highly correlated with strong *Bise* events (Figures 5, 6). This pattern forms and persists for several days after the wind event has ceased. Therefore, based on these results, a field campaign was designed to capture the boundary and temporal variation of CG1 from 20 to 22 September 2019.

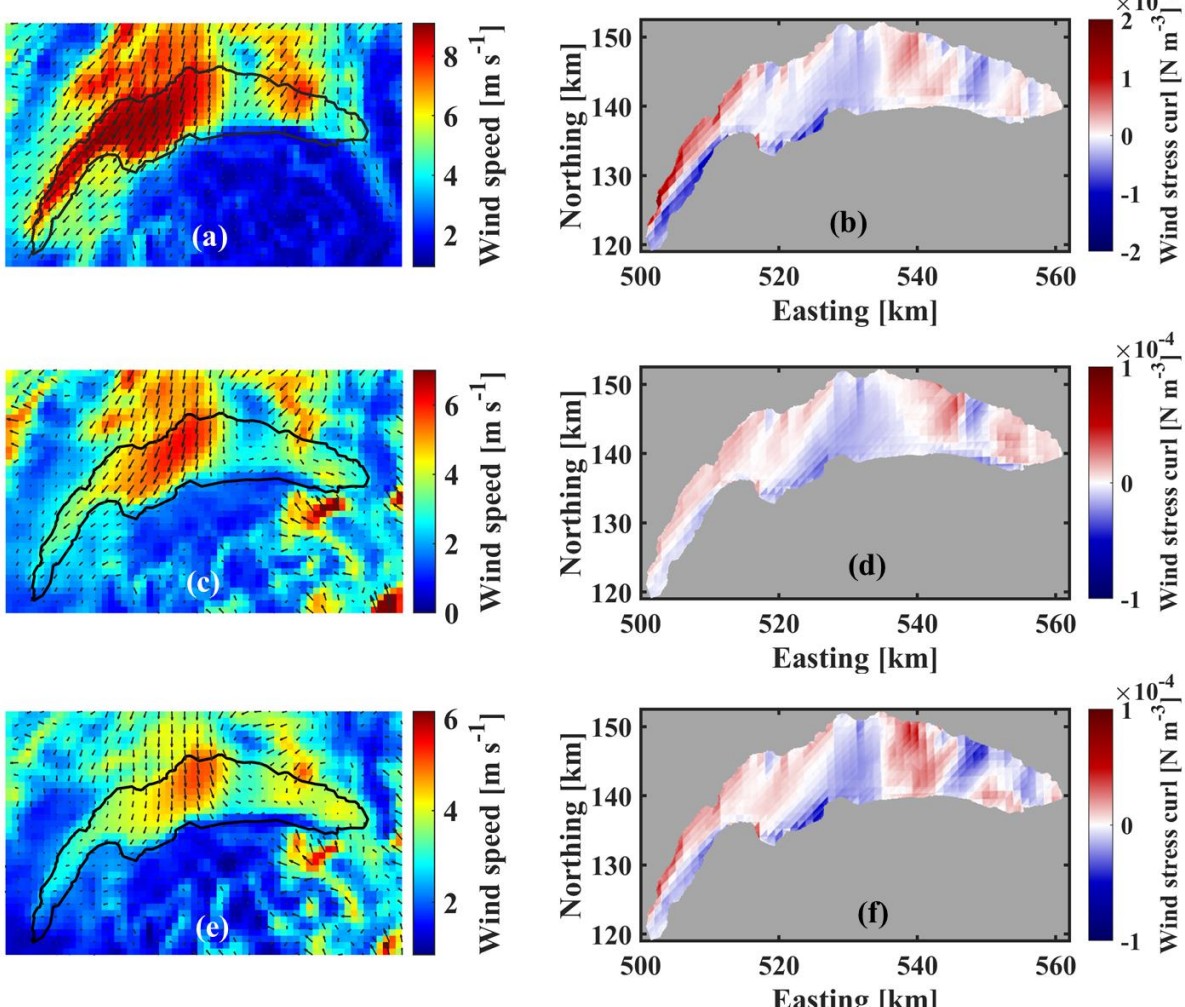

**Figure 7.** Left column: Averaged wind speed and direction extracted from COSMO data for *Bise* events. Right column: The computed wind stress curl during the corresponding *Bise* events. (a) and (b) from 17 to 20 September 2019; (c) and (d) from 21 to 23 October 2019; (e) and (f) from 18 to 23 November 2019. Colorbars indicate the range of parameters.

Four transects, T2, T2L, T2R, and T2H (Figure 1b; coordinates are given in Table S1) were chosen to investigate the 3D structure of CG1 and its evolution. A few hours after the strong *Bise* event ended on 20 September, measurements at 10 points with 1-km spacing along transect T2, from south to north, were taken. The duration of field measurements along the vertical (T2, T2L, T2R) and horizontal (T2H) transects was 2 to 3 h each. More details are given in Table S1. The measured velocity field confirmed the existence of a cyclonic circulation in the center of the *Grand Lac* (Figure 8a, b). In the proposed procedure, the center of each gyre is calculated by averaging the coordinates of the centers of the minimum OW$_N$ zones in different depth layers. To take into account the uncertainty in selecting the location of each transect, the standard deviation of the coordinates of the minimum OW$_N$ zones from the average transect was also calculated. It was ~1.8 km for September 2018.

This uncertainty was considered by taking measurements along two additional transects, T2R and T2L, that covered the low velocity core zones (almost zero) surrounding the center. The velocity field for each transect is given in Figure 8c. The 3D velocity field shows a cyclonic gyre. The gyre velocity field penetrated down to ~15 m, limited by the strong thermal stratification in September (see Figure S1 for temperature profile at SHL2 near the center of CG1). The maximum horizontal water velocity reaches ~35 cm s$^{-1}$ in the near-surface layer. The center of the cyclonic gyre can clearly be seen in all transects by the strong decrease in horizontal velocity.

In order to capture the gyre boundary and to determine the complete CG1 velocity field, 12 points with 1.5-km spacing were measured along transect T2H (see Figure 1b for location), from west to east on 22 September. For example, point 1 at the western end of the transect was clearly outside the gyre field because both the velocity magnitude and direction changed. Transect 2 (T2), from south to north was repeated. The field results (Figure 8d) confirmed the boundaries of CG1, which were identified in the SAR image (see Figure 4a). Furthermore, the velocity field shows that the nearshore (south and north) currents are much stronger than currents at the east-west boundary of CG1, indicating that the nearshore bathymetry deforms/confines the gyre flow field.

An analysis similar to that of September 2018 discussed above was carried out for September 2019, with similar results (Figure S8). Comparisons between the modeled and observed velocity fields for the September 2019 campaign are given in Figure S3, again confirming the close resemblance between the results of field observations and the numerical modeling.

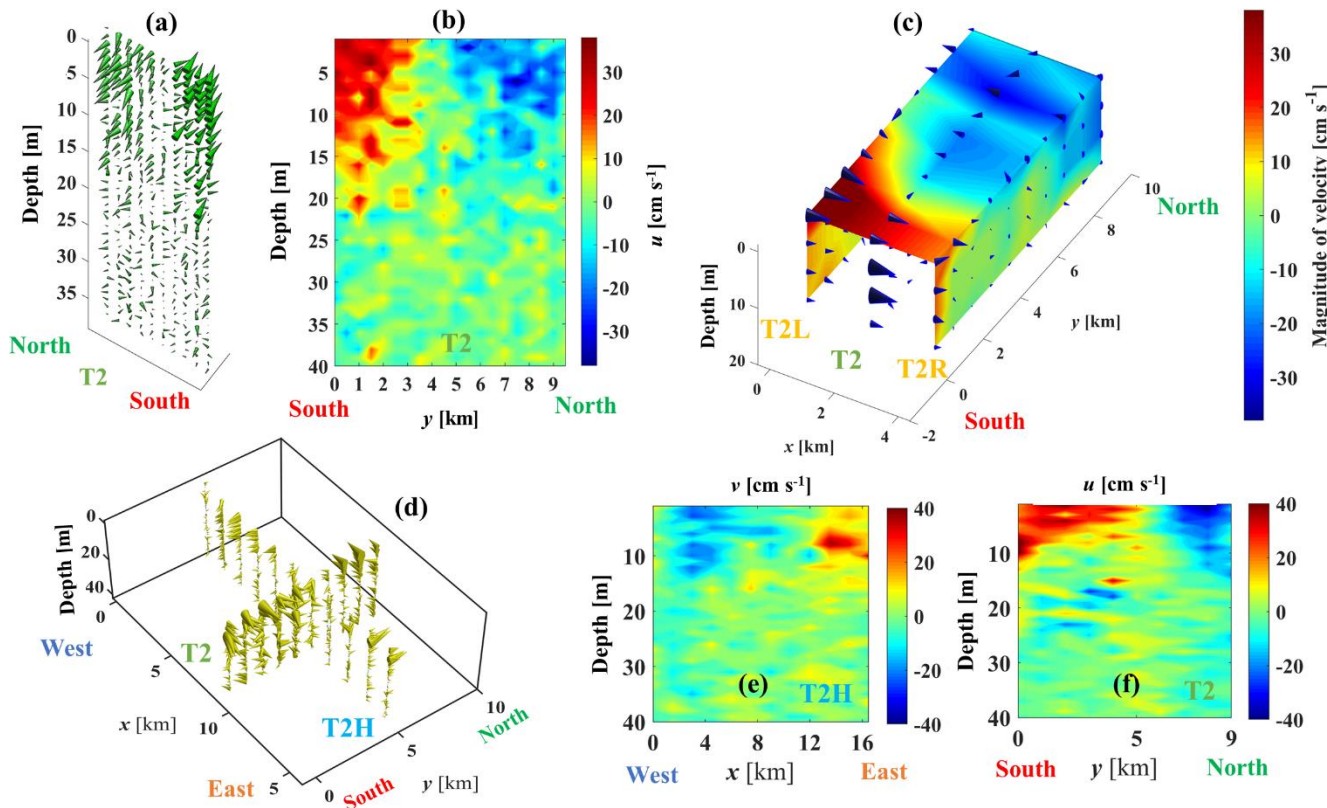

**Figure 8.** Measured current velocity field for sampling points along the selected transects (T) shown in Figure 1b for: (a) current vector profiles and (b) contour plot for 20 September 2019; (c) vectors (black arrows) and contours (see colorbar legend) for 21 September 2019; (d) current vector profiles; (e) and (f) contour plots for 22 September 2019. The coordinates of sampling points in T2 are the same in (a)-(d) and (f). The colorbars indicate horizontal velocity in cm s$^{-1}$. Positive velocities are pointing eastward for transect T2, and northward for transect T2H. Colorbars give the range of the horizontal velocity.

### 4.2.2 Observations of two- and three-gyre patterns

In most previous numerical studies on Lake Geneva, two basin-scale gyres (a dipole), located in the center of the deep *Grand Lac* basin were considered to be the main basin-scale circulation (Le Thi et al., 2012; Razmi et al., 2017), although the EOF analysis predicted three large-scale gyres (Figures 3, 4). To confirm the existence of a dipole in Lake Geneva, field measurements were conducted from 23 to 25 October 2019 and from 24 to 26 November 2019 after strong *Bise* wind events (Figure S9). However, the primary objective of the field campaign on 25 October and 25 November 2019 was to determine whether three large-scale gyres predicted by the EOF analysis (Figures 3, 4) exist in Lake Geneva.

The average wind direction during the *Bise* event in October 2019 was ~69° with a duration of 40 h (Figure 7b), and was ~47° with a duration of 94 h in November 2019 (Figures 7c, S9). For the October 2019 campaign, two transects, T2

(western part) and T3 (eastern part) (Figure 1b), which consisted of 10 measurement points with 1-km spacing, were selected following the proposed procedure, but with different transect coordinates for each month. The duration of field measurements along transects T2 and T3 was ~2 h each. To confirm the existence of the third gyre in the eastern part of lake, CG2, two transects were selected, T1 and T1H (see Figure 1b), each with eight points. The distance between the T1H measurement points was 1.5 km in order to capture the CG2 boundary observed in the SAR image (Figure 4). The duration of the field measurements along transects T1 and T1H was ~1.5 and 2.5 h, respectively. In the 3D velocity field recorded along different transects during the October campaign (Figure 9a), a dipole consisting of an anticyclonic gyre, AG, at transect T3, and a cyclonic gyre, CG1, at T2 is clearly evident. The maximum velocity at CG1 (22 cm s$^{-1}$) is stronger than at AG (15 cm s$^{-1}$), with the velocity field of AG penetrating into deeper layers. The 3D velocity field observed at T1 and T1H confirms the existence of the third cyclonic gyre (CG2) in the eastern part of Lake Geneva (Figure 10). The CG2 velocity field penetrated down to nearly 20 m and the maximum horizontal water velocity reached ~17 cm s$^{-1}$ near the surface layer. The depth of the CG2 gyre is similar to that of CG1 and the magnitude of its velocity is comparable to AG.

To further confirm the existence of the three-gyre pattern in Lake Geneva during weakly stratified months, measurements along the three transects, T1, T2, and T3 (Figure 1b), were carried out in November 2019, based on the EOF analysis for November 2018. During November, a different location for T3 was chosen because, compared to the October campaign, a different spatial pattern was observed in the EOF results (see Table S1 and Text S2). The transects T1 (eastern part), T2 (central part) and T3 (western part) were sampled on the same day, with 9, 10 and 8 points, respectively, each with a 1-km spacing. The duration of field measurements along transects T1, T2 and T3 was ~2 h each. The measured 3D velocity fields reveal two cyclonic circulations at T1 and T2 and one anticyclonic circulation at T3 (Figure 9d-g). In contrast to the October campaign, the magnitude and depth of the velocity field at CG2 and CG1 are comparable. Comparisons between the modeled and observed velocity fields for October and November 2019 campaigns show good agreement (Figures S4 and S5 in SI).

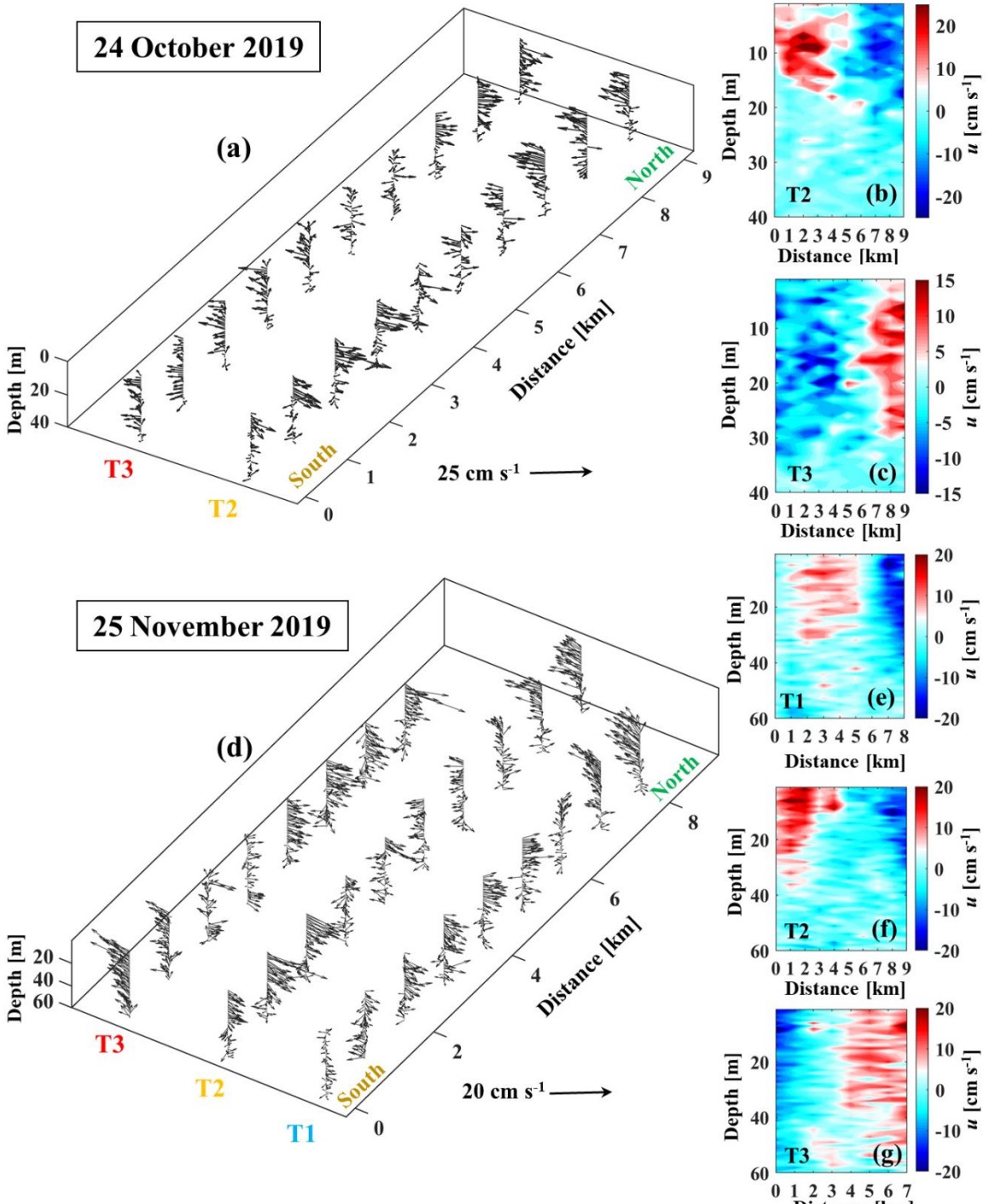

**Figure 9.** Field campaigns of October and November 2019 carried out along selected transects (T): (a) Current velocity vector fields (arrows) of gyres AG (T3) and CG1 (T2), and (b, c) contour plots of the horizontal velocity, $u$, for 24 October 2019. (d) Current velocity vector field

of gyres AG (T3), CG1 (T2) and CG2 (T1), and (e-g) contour plots of the horizontal velocity, $u$, for November 2019. The origin of the $x$-axis is in the south. The colorbars indicate horizontal velocity in cm s⁻¹. Positive velocities are pointing eastward for transects T1, T2 and T3. For transect locations, see Figure 1b.

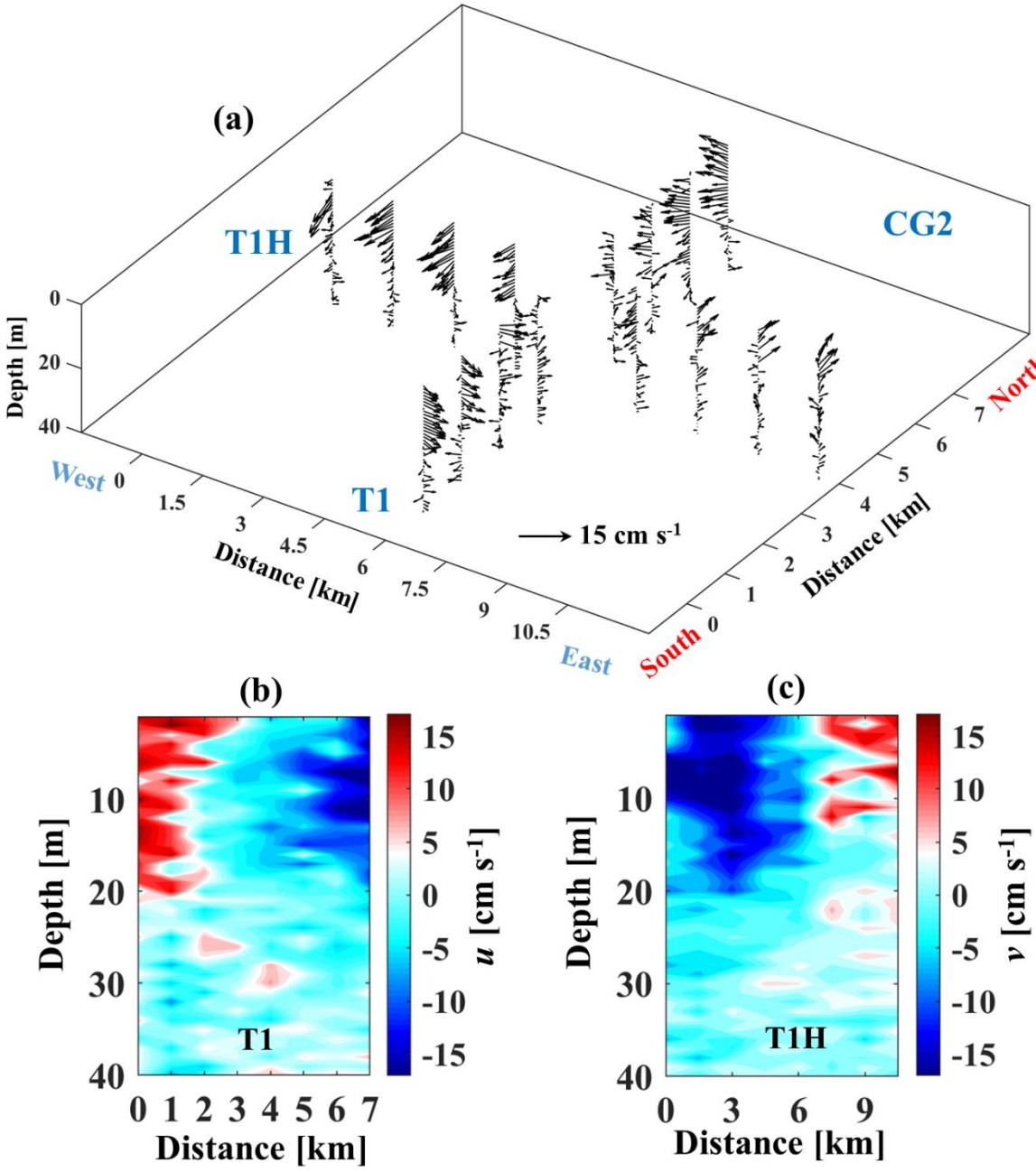

**Figure 10.** Field campaign of 25 October 2019: (a) Current velocity field of CG2 shown as vector profiles at the different stations along transects T1 and T1H. (b) and (c) Contour plots of the horizontal components of velocity field, $u$ (along T1) and $v$ (along T1H). The origin

of the *x*-axis is in the south for T1 and in the west for TH1. The colorbars indicate horizontal velocity in cm s$^{-1}$. Positive velocities are pointing eastward for transects T1, T2 and T3, and northward for transect T1H. For transect location, see Figure 1b.

## 4.3 Detecting small eddies

Small eddies have not yet been investigated in Lake Geneva. In the present study, smaller-sized coherent structures with
415 negative OW$_N$ values were detected in the vicinity of the basin-scale gyres or at coastal headlands (Figure 3). Two dominant and frequently occurring patterns of such eddies were selected in the EOF analysis for different months (Figures 11, 12). The focus here was on Submesoscale Eddies, SE1 and SE2 (see Figure 11c for location) that have a lifetime of several days, comparable to the lifetime of basin-scale gyres. In the SAR images, small eddies appear as radar-dark filaments wound into spirals (Hamze-Ziabari et al., 2022). A close similarity between the patterns observed in the SAR data and numerical results
is observed (cf. Figures 11, 12). According to the numerical results, the rotation of SE1 is cyclonic and its diameter was O(5) km during October 2019.

Such small-scale patterns cannot be adequately captured by velocity measurements at fixed points. Therefore, the ADCP was mounted on a small, boat-towed catamaran. Continuous vertical current profiles were measured along the preselected transects. SE1 was frequently observed by the proposed procedure during October 2018 and 2019 (not shown). A
425 field campaign was conducted in October 2020 to investigate the vertical structure of SE1. Two transects, T4 and T5, were chosen based on EOF results and SAR images for 2018 and 2019. The horizontal lengths of T4 and T5 were ~8.5 and 3.5 km, respectively. The spatial resolution of the measured current profiles depends on the boat's speed (1.5-2 km/h). The duration of field measurements along transects T4 and T5 was ~5 and 2 h, respectively. Velocity data were averaged every minute, which resulted in a 25-33 m spatial resolution along each transect. The numerical results and measured horizontal velocities at T4
and T5 are given in Figure 11e-h. A cyclonic circulation formed in the southwestern part of lake, and an anticyclonic circulation in the northwestern part (Figure 11d). There is a close match between the measured and modeled velocity fields (see Figure 11e-h). Field data indicate that the velocity field of SE1 extended to depths between 35-40 m, comparable to the depth of the anticyclonic gyre (AG) in this part of the lake measured during the October 2019 campaign.

Flow separation in the vicinity of headlands or baroclinicity due to favorable upwelling in the *Petit Lac* can also lead
to the formation of submesoscale eddies, such as SE2 (Figure 12). The dimensions of SE2 are generally smaller than those of SE1. However, SE2 is more frequently observed during certain months of year. For example, the signature of SE2 can be clearly observed in SAR images taken on 19 August 2018 and 7 November 2018 (Figure 12). The numerical results confirm the presence of cold cyclonic eddies with the same dimensions for the same period (Figure 12b, d). The lateral extent of SE2 is O(4 km). A field campaign on 3 September 2020 confirmed the existence of SE2. The simulated and measured velocity
fields indicate a cyclonic circulation in the northwestern part of *Grand L*ac basin of Lake Geneva (Figure 12e, f). Field data show that the SE2 velocity field can reach 20-m depth, which is 5-m deeper than the depth of the cyclonic circulation during the September 2019 campaign.

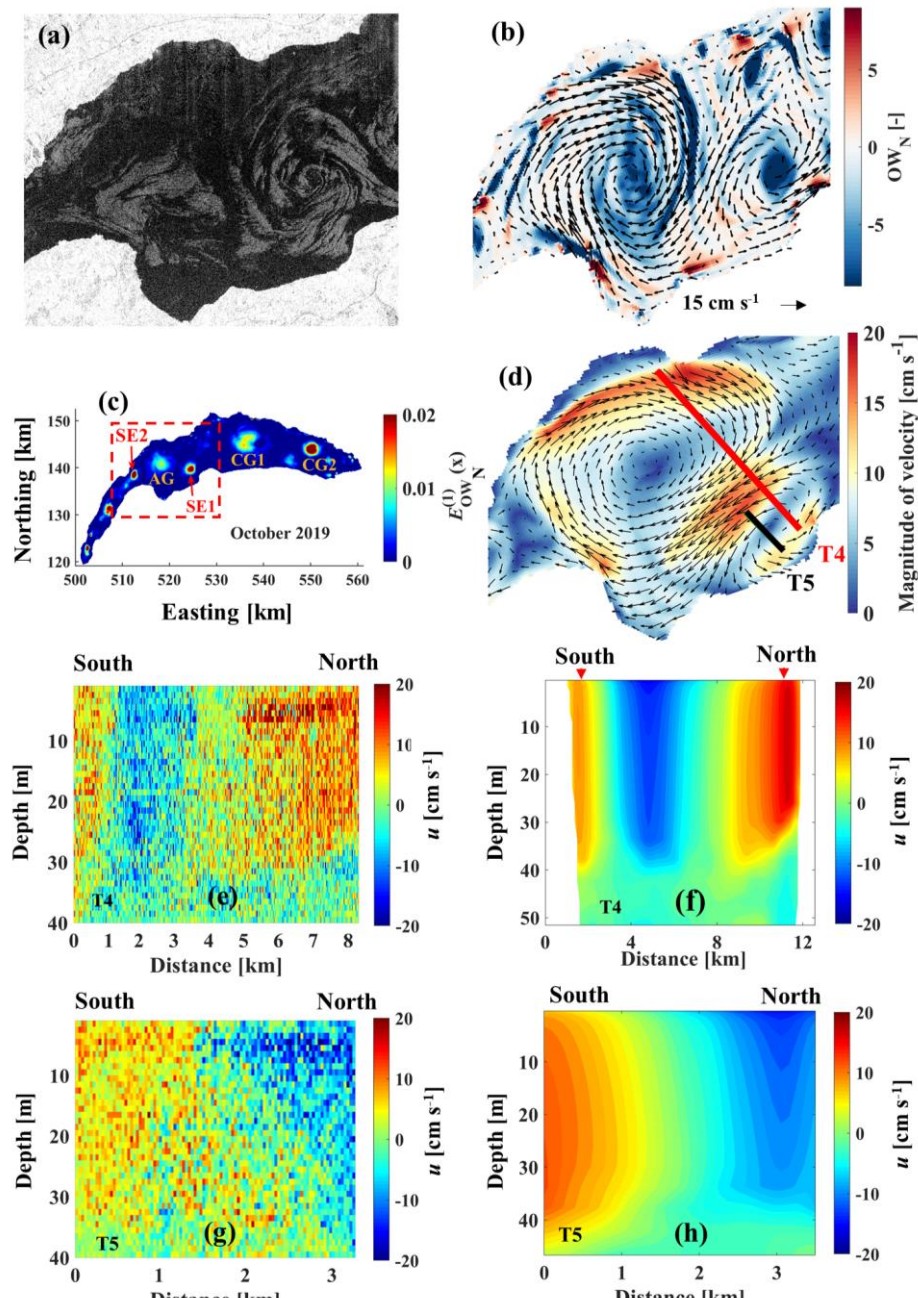

**Figure 11.** Evidence for the existence of Submesoscale Eddy 1 (SE1). (a) Remote sensing evidence (SAR images, Sentinel-1) of SE1 for 27 October 2019. (b) Corresponding modeled surface velocity fields (small black arrows show sense of rotation) and $OW_N$ values shown by colors. (c) First EOF mode of $OW_N$ for October 2019 showing large gyres AG1, CG1 and CG2 and small eddies SE1 and SE2. (d) Modeled surface velocity fields for 19 October 2020 and selected transect locations (T4, T5). (e) Measured horizontal current velocity profile along T4. (f) Corresponding modeled horizontal velocity along T4. (g) Measured horizontal current velocity profile along T5. (h)

Corresponding modeled horizontal velocity along T5. Red dashed-lined rectangle in (c) indicates the location of zoom panels (a), (b) and (d). Colorbars give the range of the parameters in the panels. Positive horizontal velocity points eastward.

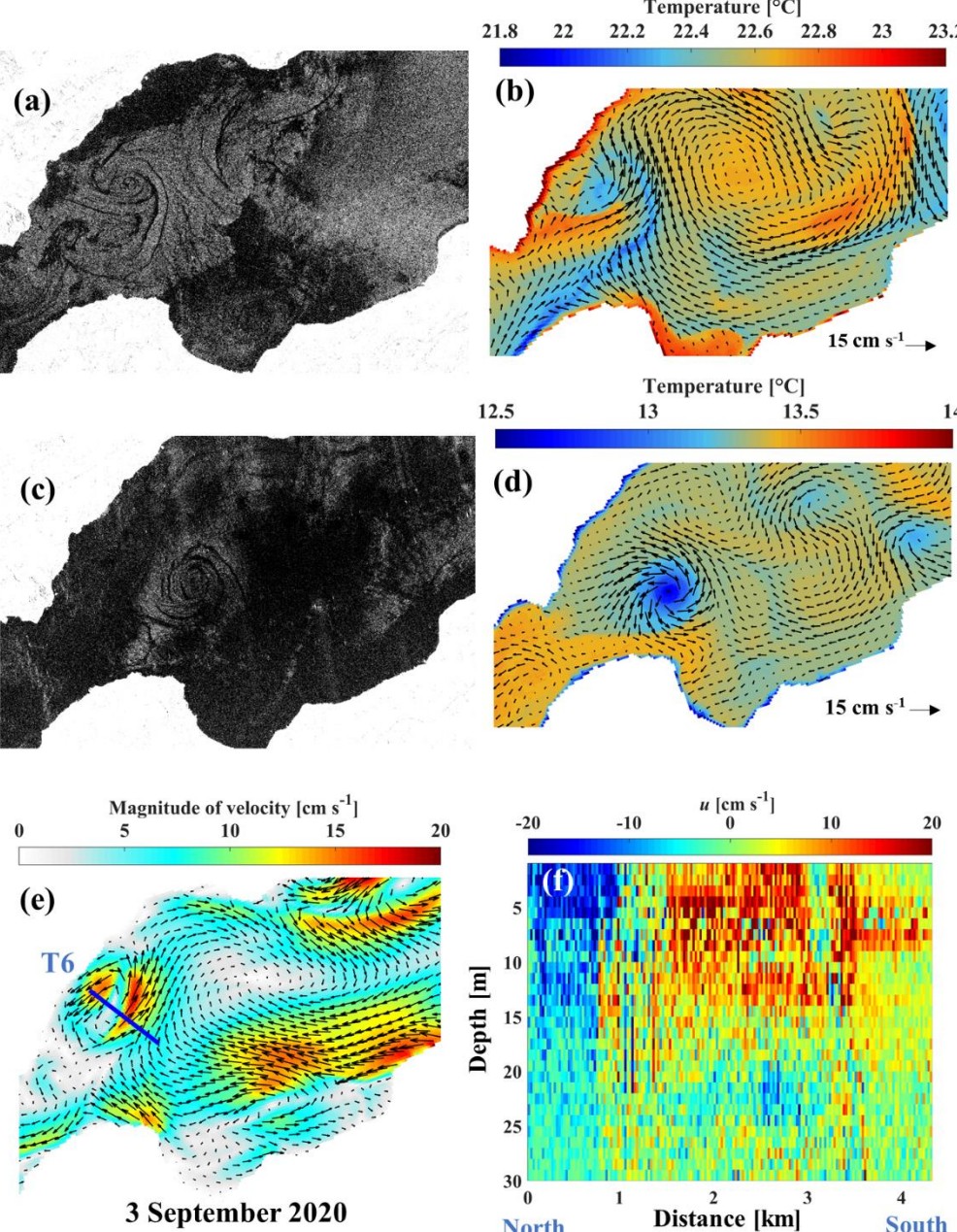

 **Figure 12.** Evidence for the existence of Submesoscale Eddy 2 (SE2). (a) Remote sensing evidence (SAR image, Sentinel-1) of SE2 for 19 August 2018. (b) Corresponding modeled surface velocity fields (small black arrows) and temperatures shown by colors. (c) SAR image of SE2 for 7 November 2018. (d) Corresponding modeled surface velocity and temperature fields for 7 November 2018. (e) Modeled surface velocity and selected transect (T6) for the field campaign of 3 September 2020. (f) Measured horizontal current velocity along T6. For

location of these panels, see Figure 11c. Colorbars give the range of the parameters in the panels. Positive horizontal velocity points eastward.

## 5 Discussion

### 5.1 Proposed procedure for detecting gyres

Large scale gyres and eddies are important transport processes in large lakes that provide for the rapid spreading of materials from the nearshore zone into the lake interior. The present study proposed a procedure for locating gyres and small eddies in a large lake and also for determining the details of their patterns. Its application to Lake Geneva has shown that a unique relationship exists between *Bise* wind forcing events and the resulting three-gyre pattern. This pattern is predictable and stable in space and time for a certain period after the forcing has ceased. The subsequent confirmation of the results by transect field measurements whose location and timing was based on the predicted pattern has demonstrated the feasibility and the robustness of the procedure. As a result, strategies for such field campaigns can be designed ensuring a high success rate in detecting gyres and eddies. Without the information provided by this procedure, it is almost impossible to carry out such field campaigns, given the complex nature of circulation patterns that exist in Lake Geneva as highlighted by the EOF results.

### 5.2 Gyre pattern: Two or three gyres?

Numerical studies have shown that large-scale gyres contribute significantly to the transport of water masses and potential pollutants from the nearshore zone into the interior of the lake (Cimatoribus et al., 2019; Reiss et al., 2020). Particle tracking used in these studies indicated a rapid spreading of these water masses over large areas of the *Grand Lac* basin within a short time, once they were caught in gyres. Previous numerical simulation studies on Lake Geneva suggested that two basin-scale gyres located at the center of the deep *Grand Lac* basin drive the main basin-scale circulation (Le Thi et al., 2012; Lemmin, 2016; Razmi et al., 2017) and can have an impact on lake bio-chemical-physical interactions (Cotte and Vennemann, 2020). However, no field measurements were carried out to confirm the existence of a two-gyre pattern reported in those studies. In the present study, transect field measurements provided detailed evidence of these two gyres.

Furthermore, transect field measurements confirmed the existence of a third gyre located in the eastern part of *Grand Lac* that is as well developed as the other two gyres. Previously, modeling had suggested that this third gyre may be important for the rapid spreading of water masses and potential pollutants brought into the lake by the Rhône River because its plume directly fed into this gyre (Lemmin, 2016). Generally, the signature of the cyclonic gyres, CG1 in the center and CG2 in the eastern part of the lake, is more pronounced than that of anticyclonic gyre, AG, in the west. The western part of lake is directly exposed to sustained strong winds, in contrast to the topographically sheltered eastern part (Rahaghi et al., 2019; Figures 5a, 7). Based on the EOF analysis in the proposed procedure, it could be established that different from previous individual gyre observations, three gyres occur regularly and have a consistent pattern. The link between the three-gyre pattern and wind

forcing was made evident. In addition, it was demonstrated that the location of gyre centers and the duration of the gyre pattern only changed slightly between *Bise* events.

## 5.3 Forcing: Importance of spatial heterogeneity

The dominant role of wind as the primary force in gyre formation, the impact of stratification and Coriolis forcing, and the importance of surface heating and cooling in driving or enhancing gyre flows have recently been highlighted (Hogg and Gayen, 2020). The spatial pattern of the net heat flux suggests that the heterogeneity between heating and cooling in the eastern part of the lake is more significant than in the western part (Figure 5b). This agrees with findings of spatial heat flux variability by Rahaghi et al., (2019). The variability may impact on the strength of gyre flows in the eastern part of Lake Geneva. Further research is needed in order to quantify the role of differential surface buoyancy fluxes on the strength of cyclonic gyres in the eastern part of lake, particularly during summertime when diurnal heating and cooling are stronger.

As shown in Figure 9, the depth influenced by the gyre field (AG) in the western part of the lake is greater than the depths influenced by CG1 located in the center and CG2, in the eastern part of the lake for the November and October campaigns. The gyre velocity is constrained by the thermocline depth, as previously discussed. The thermocline depth can also be affected by the spatial heterogeneity of atmospheric forcing and gyre motions. Forcing by wind stress, the primary source of energy for mixing the water column, is more pronounced in the western part (Figures 5a, 7). Consequently, a deeper mixed layer would be expected in the western part of the lake. The lower velocity of gyre flow in the western part of the lake can be attributed to the fact that the wind energy can penetrate into deeper layers due to a deeper thermocline, whereas it is confined by a shallower thermocline in the eastern part. As a result, the maximum velocity field of CG1 and CG2 is generally greater than AG.

In many large water bodies surrounded by complex terrain, a spatially variable wind field is one of the most important driving forces for the excitation of both horizontal and residual circulations (Nakayama et al., 2014). It was suggested, for example, that wind stress curl played a significant role in forming cyclonic/anticyclonic circulations in many lakes, such as Lake Superior (Bennington et al., 2010), Lake Michigan (Schwab and Beletsky, 2003), Lake Kinneret (Israel; Laval et al., 2003), Lake Tahoe (USA; Rueda et al., 2005) and Lake Biwa (Japan; Shimizu et al., 2007). The wind field in Lake Geneva, which is also surrounded by a complex terrain (Figure 1), can vary spatially as shown in Figure 7. The lake surface may experience both positive and negative wind stress curl in different areas as a result of such a variable wind field. Under constant positive (negative) wind stress curl, divergent (convergent) Ekman transport can lead to formation of cyclonic (anticyclonic) circulations. Several areas with positive and negative wind stress curls can be identified on the surface of the lake (Figure 7b, d, e). In general, the magnitude of positive and negative wind stress curl is greater in nearshore zones than in pelagic areas. The wind stress curl is both positive and negative in the CG1 and CG2 areas. However, it is insignificant or positive at the location of AG. Thus, Ekman pumping cannot be responsible for the formation of the anticyclonic circulation in AG located in the western part of the lake. Positive wind stress curl in the eastern part of the lake may, however, play a role in the excitation of cyclonic circulations (Lemmin and D'Adamo, 1996). Further research is required to quantify the effects of spatial variability

of surface heat flux (Rahaghi et al., 2019) and wind stress curl on the development of the three-gyre pattern observed in the field.

## 5.4 Effect of thermal stratification on the gyre velocity field

The thermal stratification of the water column, which is subject to seasonal changes, can determine the depth affected by meso- and basin-scale gyres. The temporal evolution of the gyre flow field in the vertical direction is investigated based on horizontal velocity contour profiles ($u$) measured at the center of CG1 from September to December 2019 (Figure 13). Since transect T2 (see Figure 1), located in the center of CG1, is not exposed to strong wind events, seasonal cooling can be considered as the primary factor controlling the increase in thermocline depth with time. As a result, the depth of the water column affected by
the gyre motions continuously increases from September to December, while the maximum current velocity at near-surface layers diminishes (Figure 13). The CG1 flow field was found to extend to depths of approximately 15 m in September, and 20 m in October, which corresponds to the thermocline layer depth where strong temperature drops of ~10°C and 8°C, respectively, were observed (Figure 13).

       The gyre flow field penetrated to depths of ~40 m in November and ~60 m in December. This is deeper than the depth
at which the thermocline layer began to form. For November and December, the temperature drops in the thermocline layer were ~4°C and ~3°C, respectively. Due to seasonal cooling, the thermocline strength, i.e., the temperature-depth segment connecting a point below the mixed layer and the thermocline depth, decreases from September to December. These observations indicate that the thermocline layer can act as a physical barrier that confines gyre velocity fields in the vertical direction during strongly stratified conditions. Under weak stratification conditions, the gyre velocity field can penetrate into
the thermocline layer due to the weakening of this physical barrier. Therefore, the thermocline layer is not an absolute barrier. Furthermore, the temperature profiles measured at the center and in the nearshore areas of transect T2 indicate that the thermocline can be dome-shaped (Figure 13). This suggests pelagic upwelling in the center of the gyre. However, this aspect cannot be treated in detail with the available data, and is beyond the scope of the present paper.

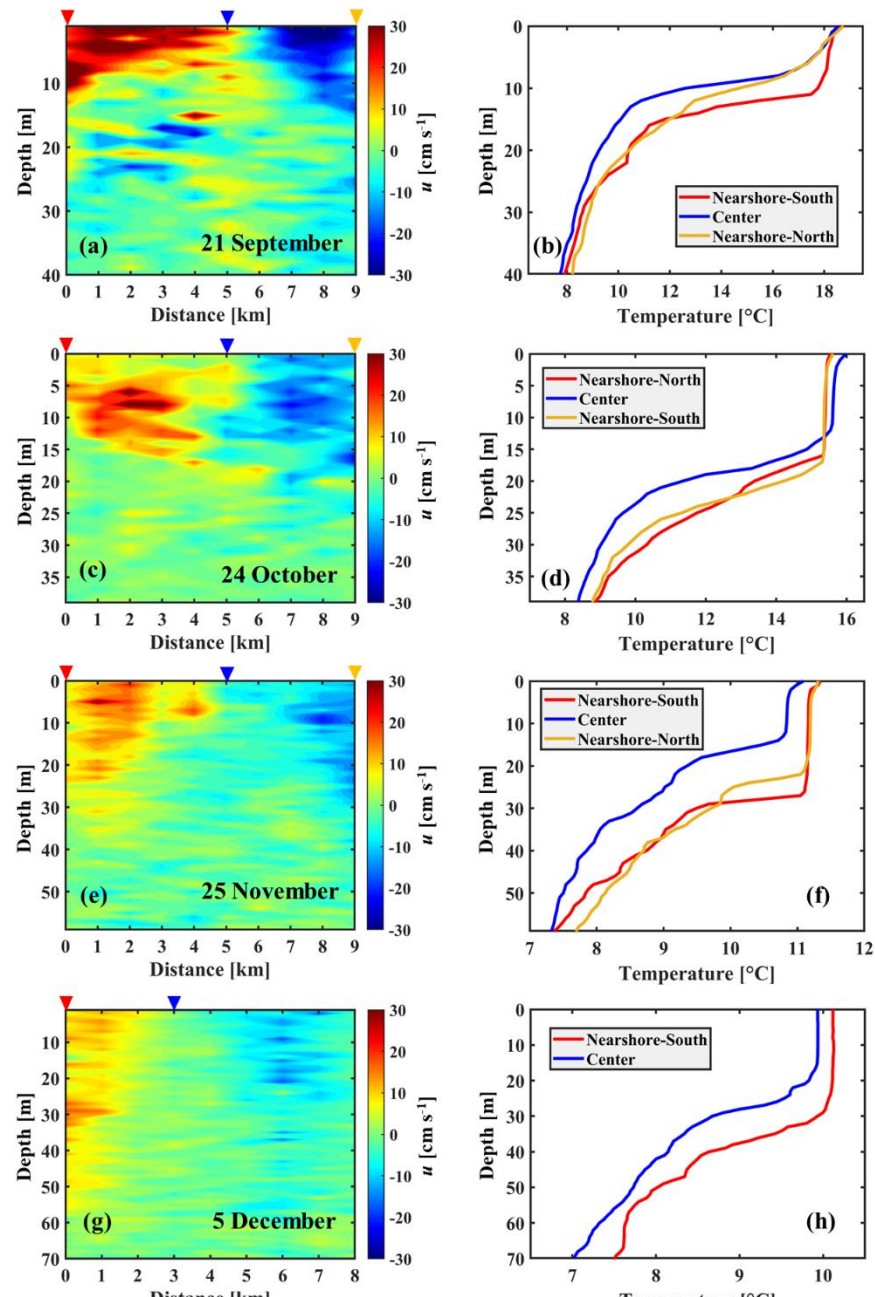

**Figure 13.** Left column: Contour plots of the measured horizontal velocity along Transect T2 (for location, see Figure 1b). Right column:
Measured temperature profiles at the locations marked by corresponding colored triangles above panels in the left column: (a) and (b) 21
September 2019; (c) and (d) 24 October 2019; (e) and (f) 25 November 2019; (g) and (h) 5 December 2019. Note that all panels in the left
column are plotted with the same range to highlight the seasonal change; the range of the *x*-axis in the right column changes with time.
Colorbars show the range of horizontal velocities.

**5.5 Effect of the Coriolis force on the gyre velocity field**

Field campaigns and remote sensing data indicate that the dimensions of the three gyre systems are larger than the internal Rossby radius of deformation for Lake Geneva (O(5 km) during summer and O(1 km) during winter). This implies that the circulation induced by external forcing is affected by the Coriolis force. To further investigate the effect of Coriolis force on the gyre flows, the vertically-averaged horizontal momentum equation can be written as (Vallis, 2017; Cimatoribus et al., 2018):

$$\frac{\partial U_h}{\partial t} = P + N + C + F - D \qquad\qquad (6)$$

where $U_h$ is the horizontal velocity field, $P$ is the acceleration induced by the barotropic and baroclinic pressure gradients, $N$ is the acceleration caused by the nonlinear (advection) terms, $C$ is the acceleration caused by the Coriolis force, $F$ is the acceleration induced by external forces and $D$ is the deceleration induced by dissipation (i.e., bulk, lateral and bottom frictions). The zonal and meridional momentum trends of the Coriolis term ($C$) are first determined. Then, the magnitude and direction

of momentum trends induced by $C$ are examined during and after the wind event.

Figure 14 illustrates the temporally-averaged horizontal momentum trends caused by term $C$ during (Figure 14a) and after (Figure 14b) the *Bise* wind event from 17 to 20 September 2019. During the *Bise*, the resultant near-surface currents (depth-averaged over 5 m) are oriented towards the north of the lake, due to Ekman transport (Figure 14a). After the wind event ceased, the three-gyre system is fully developed (Figure S3a). The Coriolis force exerts different effects on cyclonic and

560 anticyclonic circulations (Figure 14b). Due to the convergence of the AG flow field caused by the Coriolis force, a bowl-shaped thermocline can form at the center of an anticyclonic gyre. Conversely, the Coriolis effect tends to diverge in the CG1 and CG2 flow fields, leading to the formation of a dome-shaped thermocline, as confirmed by the temperature profiles measured during different months at the center of CG1 (Figure 13).

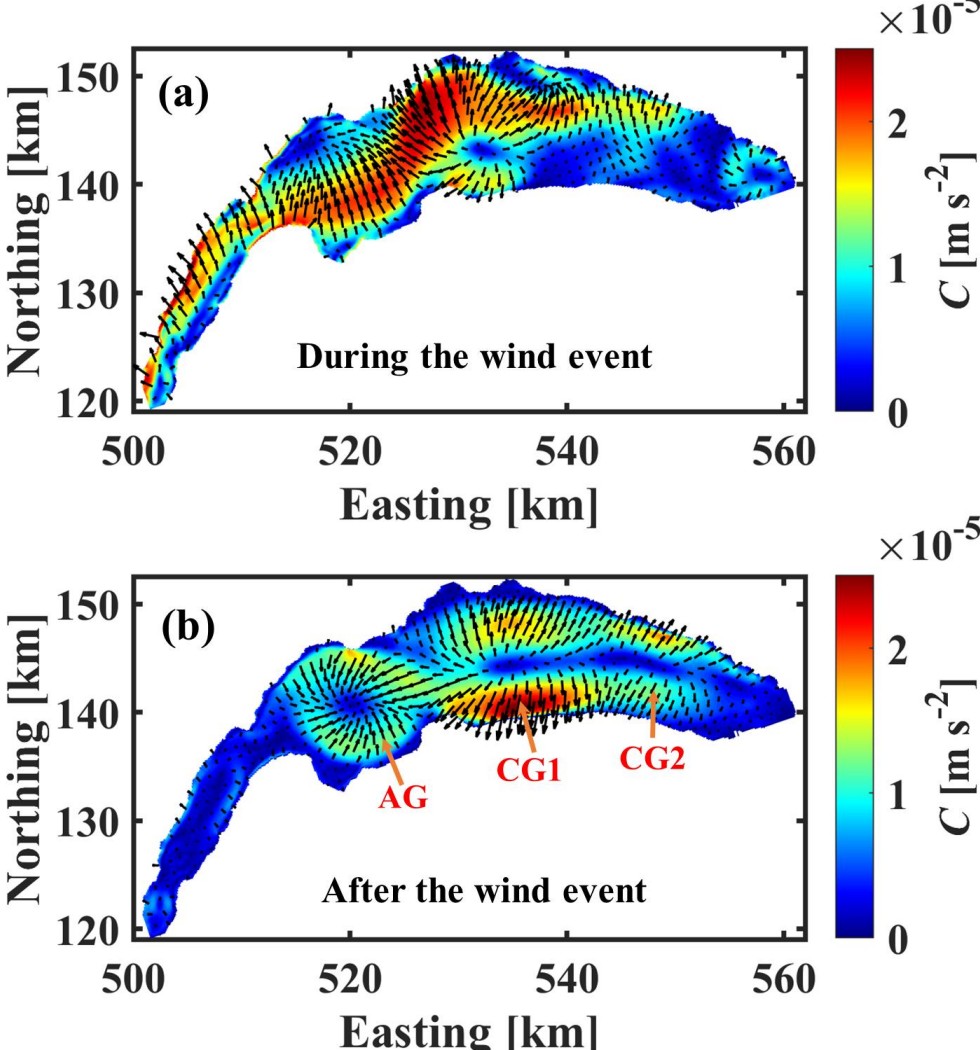

**Figure 14.** (a) Contribution of Coriolis term $C$ (strength as indicated by color contours, and orientation, by arrows) to the vertically (upper 5 m of the water column) and temporally (65 h) averaged horizontal momentum equation (Eq. (6)): (a) during and (b) after the *Bise* event that occurred in September 2019. After the wind event, convergence can be seen at gyre AG, and divergence in the CG1 gyre area.

## 5.6 Small eddies

The spatial pattern in the $OW_N$ computed for October 2019 is slightly different than that of September 2019, because a strong smaller-scale cyclonic circulation (SE1) exists near gyre AG in the western part of the lake (Figure 11). Small eddies, with approximately the size of the Rossby radius, are predicted by the numerical modeling and are important in mixing and transport of lake water masses. They were most often observed shedding from headlands, trapped between basin-scale gyres, as well as being ejected from bays (Razmi et al., 2017). Although it is beyond the scope of this study to investigate the origin of small

eddies, it was demonstrated that the proposed procedure can detect and locate them correctly, as was confirmed by transect field measurements. Small eddies can play an important role with respect to sea and ocean turbulence, stratification and primary production (Mahadevan, 2016). Generally, such small eddies are expected to be transient in nature. However, eddies SE1 and SE2 were trapped between large-scale gyres or between lake basin boundaries and gyres. They hardly lost any strength and remained active at fixed locations for several days, comparable to the lifetime of large-scale gyres.

## 6 Summary and Conclusions

In order to advance the understanding of water mass movement dynamics in large lakes, it is essential to determine the contribution of basin-scale gyres and (sub)mesoscale eddies. In particular, it is important to know to what extent the flow field generated by gyres and eddies observed during a single forcing event is "typical" and thus key to the longterm development of the lake flow field. Unfortunately, at present, a procedure that would allow identifying and tracking basin-scale and mesoscale water mass movements in a wider lake database does not exist. We therefore developed a procedure combining high-resolution 3D numerical simulations, the Okubo-Weiss (OW) parameter and EOF analysis that can provide direct evidence of the existence of cyclonic (counterclockwise) and anticyclonic (clockwise) rotating basin-scale gyres and meso-scale eddies in large lakes. Its feasibility and robustness was assessed and confirmed by field measurements taken in Lake Geneva along transects whose location and timing was based on the numerical modeling predicted pattern. The results of this study can be summarized as follows:

- The gyre flow field is characterized by a coherent pattern of the normalized Okubo-Weiss parameter, $OW_N$, in different layers of the lake, as was detected by the EOF analysis. The results showed a clear link between strong large-scale wind events and the computed spatial patterns in the first mode of the EOF analysis. The procedure allowed detection of the location of gyre centers where almost zero horizontal velocity zones indicate the occurrence of pelagic upwelling or downwelling; this has a great impact on the biological-chemical-physical development of large lake ecological systems.

- Field observations confirmed for the first time that three persistent gyres, two cyclonic gyres and one anticyclonic gyre, formed after strong *Bise* wind events, as was predicted by the proposed procedure. According to the EOF analysis, the horizontal gyre motion is mainly responsive to the wind stress, whereas the depth of the gyre flow field mainly depends on thermocline depth and strength.

- Field observations during October and November 2019 demonstrated that the depth of gyre penetration is greater in the western (anticyclonic gyre) than in the eastern part of the lake (cyclonic gyres). The spatial inhomogeneities in external forcing can lead to significant spatial variability of the gyre velocity field in the western and eastern parts of Lake Geneva.

- The proposed procedure can also detect (sub)mesoscale eddies, if the resolution of the numerical modeling grid is sufficient. These eddies were predominantly cyclonic, and their diameters were O(4-5 km). They may occur due to flow separation in the vicinity of headlands and embayments. These eddies are transient but, if trapped between larger-

scale gyres, they can remain at a fixed location for several days, and can have a lifetime comparable to basin-scale gyres. Their patterns were confirmed by field campaigns designed following the proposed procedure.

- It was demonstrated that by applying the proposed procedure, it is possible to develop strategies for carrying out detailed transect field studies on gyres with precision in time and space, which previously was not possible.

This study highlighted the significance of 3D processes in large lakes, thus indicating that 1D concepts cannot adequately describe the complex dynamics in such lakes. Additional research is required to investigate gyre/eddy-formation mechanisms and the role that eddies and large basin-scale gyres play in the interaction of biological-chemical-physical processes. Such large scale circulations can rapidly spread materials, including pollutants, entering the lake in the nearshore zone into the whole lake, and thus affect the long-term ecological system development of large lakes. Although the feasibility and robustness of

the proposed procedure was assessed in Lake Geneva, it can be applied to any large lake with a comparable database since it is based on universally valid concepts. It is a powerful tool for designing strategies for detailed field studies and will allow new types of field measurements that can contribute to advancing the understanding of large lake dynamics.

**Acknowledgments**

This research was supported by the Swiss National Science Foundation (SNSF Grant 178866). The spatiotemporal

meteorological data were provided by the Federal Office of Meteorology and Climatology in Switzerland (MeteoSwiss). We also extend our appreciation to the Commission Internationale pour la Protection des Eaux du Léman (CIPEL) for in situ temperature measurements. Water temperature profiles were collected at the CIPEL SHL2 station for 2018-2019 by the Eco-Informatics ORE INRA Team at the French National Institute for Agricultural Research (SOERE OLA-IS, INRA Thonon-les-Bains, France).

**Data availability**

The SAR images used in this study are based on Sentinel-1 raw data, which are made available by the ESA and can freely be downloaded from the ESA's Sentinel data hub (https://scihub.copernicus.eu/). The three-dimensional model used in this study is based on the MIT General Circulation Model (MITgcm, http://mitgcm.org/, https://doi.org/10.1029/96JC02775), which is publicly available. The in situ data and numerical configurations supporting the findings of this study are available online at

630 https://zenodo.org/record/7018567#.YwX7fBxBxPZ.

**Author contributions**

SMHZ planned the field campaign; SMHZ, FS, and MF performed the measurements; SMHZ analyzed the data; SMHZ wrote the manuscript draft; DAB and UL reviewed and edited the manuscript.

## Competing interests

The authors declare that they have no conflict of interest.

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
