# Peer review of "Basin-scale gyres and mesoscale eddies in large lakes: A novel procedure for their detection and characterization, assessed in Lake Geneva"

_Geoscientific Model Development, 2022_

## Author Comment (AC2)

RESPONSES TO REVIEWER 1

We thank the reviewer for the constructive comments and suggestions and respond to them below in blue type.

This paper provides essential information to the researchers in the same study field, particularly scientists interested in mass transport and water quality in an enclosed basin. The authors conducted significant field observations with respect and completed further analysis by using numerical simulations. However, there are some incoherences in the manuscript. For example, the authors mention the importance of the Coriolis effect on a gyre, but there is no detailed explanation and discussion. Also, the authors need to add more literature on a gyre in an enclosed basin.

Please see responses below.

Please see more details below. As mentioned above, this study could provide significant contributions after revising the manuscript. Therefore, I recommend a substantial revision of the manuscript.

1) Is only one meteorological station available despite the huge lake size?

In the vicinity of the lake, there are other meteorological stations (on land). The Buchillon station was chosen because it is located in the nearshore zone of the lake. Please note, however, that the data from this station are only used to confirm that the COSMO data (described in Section 2.2, with references), which include meteorological variables on a 1 km × 1 km grid over the entire lake (and surroundings), are realistic and can be relied upon for forecasting purposes and for the data analysis.

2) Why is there only one transect in the eastern area?

In this study, we focused on the center and western part of the lake because numerical results indicated stronger gyral flow fields in those areas. However, the eastern area was measured two times during the October and November campaigns. Transect T1H was not measured during the November campaign because the main objective of that campaign was to measure all three gyres within one day, and it was physically impossible to cover both horizontal and vertical transects with the single boat available to our team.

3) In the Materials and Methods, the authors need to mention more details about the field observations, such as the observation period.

The observation period was added to Table S1. The duration of each field campaign is now given in the text.

Modifications were made at L345-346, L383-384, L386-387, L399 and L427-428.

4) In lines 291 to 293, the authors mentioned, "the three-gyre pattern in the first mode is predominantly controlled by the spatial and temporal variations of wind stress". However, there is only one meteorological station. How did the authors investigate the spatial effect of wind on a gyre?

Section 4.1.3 already addressed this comment. We did not use any meteorological station data to find the link between external forces and the three-gyre pattern; see also our response to comment 1. To investigate the spatial effect of wind on a gyre, we use meteorological parameters obtained from the validated COSMO data which covers the full lake surface with a 1-km resolution (see Section 2.2 for details on COSMO). Cosmo data are also used to drive our model.

We added L291-294 for clarification.

5) Spatially variable winds are one of the major driving forces in many lakes and bays, such as Lake Michigan [Schwab and Beletsky, JGR, 2003], Lake Kinneret [Laval et al., L&O, 2003], Lake Tahoe [Rueda et al., JGR, 2005], Lake Biwa [Shimizu et al., L&O, 2007], and Tokyo Bay [Nakayama et al., JGR, 2014]. Could you provide any information on the spatial variation of the wind in Lake Geneva? Also, please add more discussion about the effect of wind stress curl on the occurrence of surface-layer circulation.

The averaged wind field and wind stress curl during the studied events are presented in new Figure 7. This is now further discussed in Section 5.3 (L503-519). We added the suggested references to the revised manuscript

6) The authors mention, "Coriolis force plays an important role in the formation of gyres since the width of the lake is much larger than the internal Rossby radius of deformation". I agree with it. Indeed, the authors included the Coriolis effect in numerical simulations. However, there is no investigation and discussion about the Coriolis effect on a gyre. For example, in the northern hemisphere, the vertical velocity at the gyre centre is positive when the wind curl is positive. However, there is a criterion that the vertical velocity changes from negative to positive, depending on whether the Ekman pumping is dominant. Could you add more explanation about the Coriolis effect on a gyre in the Results and Discussion?

We added a new section (Section 5.5) and a new Figure 14 to the revised manuscript to address this comment.

7) In Figures 7 and 8, the arrows need to be improved as it is difficult to understand what they show.

These figures have been improved; now Figures 8 and 9 in the revised version.

8) In lines 513 to 516, the authors mention the importance of upwelling and downwelling on large lake ecological systems. Only the authors describe the upwelling phenomenon in line 415 in the Results (Figure 10). There is no detailed explanation for it.

We describe the upwelling phenomenon at the center of CG1 by providing field observations. Further discussion can be found in new Sections 5.4 and 5.5. Please note, however, investigating pelagic upwelling or downwelling induced by cyclonic/anticyclonic gyres is beyond the scope of the present study.

9) The paper includes vital information and outcomes. However, it is not easy to follow how the authors obtain the consequences. Could you restructure the study contents to let readers efficiently understand the significance of the study?

We are not sure what is meant by "consequences." The primary objective of this study is to present a new procedure based on numerical modeling results. Every effort was made to structure the paper logically by first explaining, step-by-step, the proposed procedure, as outlined in flow chart Figure 2. We then give examples of field measurements taken on different dates that were planned and successfully executed based on the model predictions of the procedure. These campaigns demonstrated the reliability and robustness of the procedure under changing stratification. We added two new sections describing Coriolis and stratification effects on gyre flow fields, and gave more information about the spatial patterns of the wind field and wind stress curl over the lake surface during the study period, as suggested by the reviewer. Please note, however, that the focus of this paper is on the procedure to detect gyre patterns, not on the physics of gyre formation.

---

## Author Comment (AC3)

RESPONSES TO REVIEWER 2

We would like to thank the reviewer for the constructive comments and suggestions and respond to them below in blue type.

The study was carried out via comprehensive methods, including numerical models, remote sensing observations, and field observations. It made important contributions to the studies on (sub)mesoscale hydrodynamic motions and bio-geo-chemical process, and it also provided a successful example for how numerical model can be used to instruct the design of field campaigns.

Except for some minor issues in the text and figure descriptions, more detailed descriptions about the decomposition of OW parameter should be provided in the method and supplied in the results. Also, the effect from Coriolis force and thermal stratification should be further discussed if that is still within the scope of the study.

Please see responses below.

I recommend the manuscript should be accepted subject to minor revisions. Please find my specific comments below:

Abstract: the abstract contains too much introduction. More focus should be put on the major results you found via applying the numerical model and in field observations.

The Abstract was modified.

Line 88-89: this sentence is not necessary.

Authors believe that this sentence is necessary there to alert reader that there is an SI.

More discussion about the effect from Coriolis force and seasonal stratification on the gyres' size, lifetime, and boundary are required.

Two new Sections 5.4 and 5.5 and Figures 13 and 14 were added to the revised manuscript. Please note, however, that the focus of this paper is on the procedure to detect gyre patterns, not on the physics of gyre formation.

The meteorological data used to drive the numerical model were from the atmospheric model, but the wind information showed in Fig. 5 and used to identify the event was from Buchillon field station. Have you compared the model input with the realistic wind data? How is the spatial variation in the wind field? The statement or comparison figure are required to clarify that.

The data associated with the Buchillon field station were only used to demonstrate that the COSMO data are realistic and can be relied upon for forecasting purposes and data analysis. As shown in Figures 5a-d and as discussed in Section 4.1.3, the temporal and spatial variations of both wind stress and heat flux extracted from the COSMO data were used in the EOF analysis, not data from the Buchillon field station.

Modifications were made (L291-294). Furthermore, the averaged wind speed and direction during the three campaign periods based on the COSMO data are presented in new Figure 7 in order to show the spatial variation of the wind field.

Fig. 2: Describe the sources of inset images in the figure.

This information was added to the figure caption.

Line 230: Is there any specific reason for choosing September? Due to the availability of Satellite data? Or thermal stratification is vanishing in this month?

The studied months represent two different stratification conditions: strongly stratified (September and October) and weakly stratified (November and December) conditions. The thermocline layer is relatively strong in September, at a depth of approximately 10-15 m, and the epilimnion consists of a shallow mixed layer. These conditions are ideal for the formation of cyclonic gyres in lakes.

More details about the effect of thermal stratification were added to new Section 5.4.

Line 248-249: Why? The combine effect of Coriolis force?

The Coriolis effect results in flow divergence and as a result, upwelling will occur at the center of the cyclonic gyre.

A detailed discussion is given in new Section 5.5. We also added temperature profiles to new Figure 13 in order to provide evidence for pelagic upwelling.

Fig. 3: What do the percentages in the figure represent?

The values given in Figure 3 indicate the percentage of the observed spatiotemporal changes to the monthly $OW_N$ values (here September). For example, the first spatial mode in the near-surface layers accounted for approximately 56% of the total observed spatiotemporal changes in the monthly changes in $OW_N$ values, whereas the second mode only contributed 23%.

Modifications were made at L245-248.

Fig. 5: Are the values in (a) and (c) integrated over a specific time range? Can you give more explanation about how you decomposed OW parameter? In (e) and (f), the information is blur here. Why OW parameter is negative when Pown is positive? Does that mean Eown and Pown always have opposite signs?

The values in (a) and (c) represent the first mode of EOF results corresponding to the wind stress and the net heat flux for September 2018 obtained from COSMO data.

According to the Eq. (5), the OW values were decomposed into the basis function ($E_{OW}^k$) and the principal component time series as:

$$OW(X, t) = \sum_{k=1}^{N} E_{OW}^k(X) P_{OW}^k(t) = E_{OW}^{(1)}(X) P_{OW}^{(1)}(t) + E_{OW}^{(2)}(X) P_{OW}^{(2)}(t) + \cdots$$

We only kept the negative values of OW because, in the presence of gyres, the terms on the right side of Eq (5) have to be negative in order to represent elliptic regions. For better visualization, we presented only the negative values associated with $E_{OW}^{(1)}(X)$ and $E_{OW}^{(2)}(X)$, which indicated a clear gyral pattern, as defined in Section 3.1. The positive values of $E_{OW}^{(1)}(X)$ and $E_{OW}^{(2)}(X)$ are given in new Figure S2, where it can be seen that the gyral patterns are insignificant compared to the negative values. Therefore, the reviewer's interpretation is correct, i.e., $E_{own}$ and $P_{own}$ always have opposite signs.

Figure S2 and L249-253 have been added and Figure 5 was modified.

Line 404: Is that because the spatial resolution of the field measurement is not fine enough?

Since SE1 and SE2 are located in the nearshore regions, boundary currents may be mistaken for eddies. We increased the field resolution to ensure that the measured velocity field represents a cyclonic circulation in order to avoid such misinterpretations.

Line 423: Did you record the lifetime of them?

We were not able to record the lifetime based on field measurements due to frequent short (few hours) wind events, which prevented us from carrying out field measurements (we can only measure reliably and safely under calm conditions).

Fig. 9: description of panel (h)?

Added.

Line 497-499: You have said that in the result part.

These lines were removed from the result section.

---

## Referee Report (RR1)

The authors have well addressed my concerns in the first round by clarifying the effect of Coriolis force and seasonal stratification, and explaining the meteorological inputs used in the model. I only have a few comments below.

(1) L132: Corresponding to my previous comment (6), you can specify the time of weakly stratified and strongly stratified seasons here.

(2) L247: "The remaining modes contain less than 5% of the total variance." Over all the layers?

(3) Figure 13 and section 5.4 well clarified the effect of thermocline depth.

(4) In terms of the length of the article, 5.5 should be made more concise, and 5.6 can be left out or moved to the supporting information since it is separated from the main messages.

(5) L579-584 can be left out or combined into Introduction.

---

## Author Response (AR2)

We would like to thank the reviewer for the constructive comments and suggestions and respond to them below in blue type.

The authors have well addressed my concerns in the first round by clarifying the effect of Coriolis force and seasonal stratification, and explaining the meteorological inputs used in the model. I only have a few comments below.

(1) L132: Corresponding to my previous comment (6), you can specify the time of weakly stratified and strongly stratified seasons here.

The corresponding months were added (L132-133).

(2) L247: "The remaining modes contain less than 5% of the total variance." Over all the layers?

The remaining modes contain less than 5% for layers 0-30 m. The depth range is now specified in L248.

(3) Figure 13 and section 5.4 well clarified the effect of thermocline depth.

We are glad you agree.

(4) In terms of the length of the article, 5.5 should be made more concise, and 5.6 can be left out or moved to the supporting information since it is separated from the main messages.

Section 5.6 was removed. Certain pertinent points, however, were put into section 4.3 (L444-448). Since L549-551 is already mentioned in L102-104, we removed it from Section 5.5 and made further modifications.

(5) L579-584 can be left out or combined into Introduction.

We removed L579-584. We modified the Introduction (Lines 84-86) and Conclusion (Lines 569-570).